



# Temporal variability in detrital $^{10}$Be concentrations in large Himalayan catchments

Elizabeth H. Dingle[1], Hugh D. Sinclair[1], Mikael Attal[1], Ángel Rodés[2], and Vimal Singh[3]

[1]School of GeoSciences, University of Edinburgh, Drummond Street, Edinburgh, EH8 9XP, United Kingdom
[2]Scottish Universities Environmental Research Centre (SUERC), Rankine Avenue, Scottish Enterprise Technology Park, East Kilbride, G75 0QF, United Kingdom
[3]Department of Geology, University of Delhi, Delhi 110007, India

*Correspondence to:* Elizabeth Dingle (elizabeth.dingle@ed.ac.uk)

**Abstract.** Accurately quantifying sediment fluxes in large rivers draining tectonically active landscapes is complicated by the stochastic nature of sediment inputs. Cosmogenic $^{10}$Be concentrations measured in modern river sands have been used to estimate $10^2$-$10^4$ year sediment fluxes in these types of catchments, where upstream drainage areas are often in excess of 10,000 km$^2$. It is commonly assumed that within large catchments, the effects of stochastic sediment inputs are buffered such that $^{10}$Be
concentrations at the catchment outlet are relatively stable in time. We present eighteen new $^{10}$Be concentrations of modern river and dated Holocene terrace and floodplain deposits from the Ganga River near to the Himalayan mountain front. We demonstrate that $^{10}$Be concentrations measured in modern Ganga River sediments display a notable degree of variability, with concentrations ranging between ~9,000-19,000 atoms g$^{-1}$. We propose that this observed variability is driven by two factors. Firstly, by the nature of stochastic inputs of sediment (e.g. the dominant erosional process, surface production rates, depth of
landsliding, degree of mixing) and, secondly, by the evacuation timescale of individual sediment deposits which buffer their impact on catchment-averaged concentrations. Despite intensification of the Indian Summer Monsoon and subsequent doubling of sediment delivery to the Bay of Bengal at ~11-7 ka, we also find that Holocene sediment $^{10}$Be concentrations documented at the Ganga outlet have remained within the error of modern river concentrations. We demonstrate that in these systems, sediment flux cannot be simply approximated by converting detrital concentration into mean erosion rates and multiplying
by catchment area as it is possible to generate considerably larger volumetric sediment fluxes whilst maintaining comparable average $^{10}$Be concentrations.

## 1  Introduction

The quantity of sediment exported from large mountainous catchments is a fundamental control on downstream river morphology (Sinha and Friend, 1994; Dade and Friend, 1998; Church, 2006; Allen et al., 2013), the advance and retreat of coastlines
(Syvitski et al., 2005) and the growth of deltas (Orton and Reading, 1993; Goodbred and Kuehl, 1999; Galy et al., 2007). How sediment flux varies over thousand year times scales reflects changes in upstream landscape evolution which is set by climatic and tectonic conditions in active orogenic settings (Whipple and Tucker, 2002). Quantification of sediment flux from large, tectonically active catchments is challenged by the nature of the river channels (e.g. size and access), the stochastic nature of




sediment inputs (Benda and Dunne, 1997; Kirchner et al., 2001), and highly variable water discharge regimes (e.g. Collins and Walling, 2004; Singh et al., 2005; Gitto et al., 2017). Constraining sediment fluxes at intermediate timescales of $10^2$-$10^4$ years has been significantly improved through the development of detrital $^{10}$Be cosmogenic radionuclide (CRN) analysis (e.g. Brown et al., 1995; Granger et al., 1996; Niedermann, 2002; Kirchner et al., 2001; Vance et al., 2003; Von Blanckenburg, 2005). The

concentration of $^{10}$Be recorded in quartz-rich river sediments is assumed to reflect the rate of upstream landscape lowering, assuming steady-state denudation averaged over the entire upstream catchment. Based on this approach, catchment-averaged denudation rates can be calculated, and converted into CRN-derived sediment fluxes which are typically averaged over hundred to thousand year timescales (Kirchner et al., 2001; Lupker et al., 2012) where these timescales are a function of the landscape erosion rate (i.e. the time taken to erode to a depth equivalent to the cosmic ray attenuation length in that landscape) (Lal,

1991).

Sediment production, delivery and transport out of large mountain catchments is heavily influenced by stochastic inputs such as hillslope mass wasting generated by earthquakes or intense storms, or glacial lake outburst floods (Benda and Dunne, 1997; Hovius et al., 2000). In small catchments that are susceptible to such events, stochastic controls on sediment release may significantly perturb the $^{10}$Be signal measured in sediment samples at the catchment outlet (Niemi et al., 2005; Yanites

et al., 2009; West et al., 2014). In particular, deep-seated landslides excavate sediment from depths greater than the attenuation length of cosmic rays. This addition of $^{10}$Be-poor landslide material dilutes $^{10}$Be concentrations recorded in fluvial sediments sampled at the catchment outlet (Niemi et al., 2005; West et al., 2014) resulting in an over-estimation of the long-term erosion rate (Yanites et al., 2009). The timescales over which these stochastic inputs influence downstream $^{10}$Be concentrations is related to the time taken to evacuate the sediment input from the impacted reach, and also depends on patterns of intermediate

sediment storage and release (recycling) upstream of the sampling locality (Granger et al., 1996; Yanites et al., 2009; Blöthe and Korup, 2013; Scherler et al., 2014; Schildgen et al., 2016). However, even in regions dominated by high rates of landslide occurrence, it is commonly assumed that given sufficiently large catchment areas and sufficient sediment mixing, the imprint of mass wasting processes on $^{10}$Be concentrations measured at the outlet should be negligible (Niemi et al., 2005; Yanites et al., 2009).

The gross sediment flux from the Himalaya is the largest out of any mountain range on the planet and provides fertile soils for ~10 % of the global population. The vast majority of this sediment flux is sequestered in the Indus and Ganga-Brahmaputra delta and submarine fans (Lupker et al., 2011). Sediment volumes in the Ganga-Brahmaputra delta imply that overall sediment flux from these two major Himalayan river systems has halved since the early Holocene, which has been linked to a reduction in monsoon rainfall since this time (Goodbred and Kuehl, 2000; Fleitmann et al., 2007). Our current

understanding of how sediment flux from tributaries of the Ganga River into the Himalayan foreland basin varies is primarily from suspended sediment and detrital $^{10}$Be concentration data collected over the last 20 years (Ghimire and Uprety, 1990; Jha et al., 1993; Sinha and Friend, 1994; Vance et al., 2003; Andermann et al., 2012; Lupker et al., 2012). Suspended sediment data are generally based on a single daily measurement and are difficult to scale up spatially and temporally. Under these circumstances, $^{10}$Be concentrations in modern river sands can be used to generate sediment flux estimates with the advantage

of temporal and spatial averaging. However, substantial variations in $^{10}$Be concentrations from repeat river sand samples at the





catchment outlets of major Himalayan rivers have been documented (Vance et al., 2003; Lupker et al., 2012). Concentrations measured on the Ganga River close to the mountain front (near Rishikesh) vary from 9.2±1.0 to 19.5±4.1 ×$10^3$ atoms g$^{-1}$ over a 13 year time period based on three samples (Vance et al., 2003; Lupker et al., 2012); at the Kosi River near Chatara, measurements vary between 26.7±3.4 to 54.4±2.9 ×$10^3$ atoms g$^{-1}$ for three samples collected in August 2007 and November 2009, respectively (Lupker et al., 2012). Measurement errors on Ganga River samples record a $1\sigma$ of around 10-20 % of the measured concentration, whereas the measured variability from the repeat samples is >100 %. Similar observations were made along main stem samples on the Yamuna River, where discrepancies of up to ∼60 % between samples were observed (Scherler et al., 2014, 2015). This degree of variability could suggest that stochastic controls on sediment release may influence the $^{10}$Be signal, yet this is at odds with previous modelling and analysis of large catchments which has proposed that catchments of this size should be buffered against variations in detrital $^{10}$Be concentrations induced by individual hillslope events (Niemi et al., 2005).

Well preserved and dated river terraces (Srivastava et al., 2003, 2008; Sinha et al., 2010; Wasson et al., 2013) associated with the Ganga River in the west Ganga Plain present a unique opportunity to test for variations in $^{10}$Be concentrations in both ancient and modern fluvial sediments at the Himalayan mountain front. The half-life of $^{10}$Be (∼1.394 Myr) implies that any post-burial decay during the last 0.01 Myr is minimal and can be accounted for, making it the ideal technique for this approach. We analyse eighteen samples of river sands from near the outlet of the Ganga River as it crosses the mountain front. Samples are taken from modern river gravel bars, recent sand deposits of the 2013 Alaknanda floods (Dobhal et al., 2013; Durga-Rao et al., 2014; Devrani et al., 2015), and dated terrace and floodplain deposits ranging in age from ∼200 to 23,500 years. Using these data, we evaluate the short-term variability in $^{10}$Be concentrations and test for longer-term changes that are expected to reflect variations in the strength of the Indian Summer Monsoon (Sirocko et al., 1993; Gupta et al., 2005; Fleitmann et al., 2007; Clift et al., 2008; Dixit et al., 2014). Motivated by the results, we examine the impact of stochastic inputs of sediment from the upstream mountain catchment on $^{10}$Be concentrations close to the mountain front (herein referred to as the Ganga outlet). We conclude by combining field observations, data and numerical analyses results to synthesise potential drivers of CRN concentration variability in large tectonically active catchments.

## 2 Study area and context

The Ganga River is a glacially-fed perennial river rising in the High Himalaya (Fig. 1). The Ganga has two major tributaries, the Bhagirathi and Alaknanda, which join near the village of Devprayag. Further downstream, the Ganga flows through the eastern end of the Dehra Dun, an intermontane valley in the Sub-Himalaya, prior to passing through the Mohand Anticline, exiting the mountains at Haridwar before reaching the Ganga Plain (Fig. 1). The Ganga catchment is characterised by a number of broad geomorphic process domains, which can be related to the distribution of tectonic structures, topographic relief and climatic influences which vary spatially across the catchment (Fig. 2).

Upstream of the mountain front, down cutting by the Ganga River has left behind a series of strath terraces cut into Lesser Himalayan or Siwalik rocks, and cut and fill terraces in Quaternary alluvial fan deposits (Sinha et al., 2010). A number of these



terraces have been dated using optically stimulated luminescence (OSL) to reveal terrace ages of up to ∼14 ka (Sinha et al., 2010). During the transition from the Late Pleistocene to the Holocene, an intensification of the ISM is observed in a number of proxy records (Goodbred and Kuehl, 2000; Fleitmann et al., 2003; Dixit et al., 2014), which is believed to have driven a period of intense fluvial incision across much of the Himalaya (Sinha et al., 2010; Dixit et al., 2014). Erosion of pre-Holocene

sedimentary records during this period of intensified monsoon is proposed as one mechanism to explain the notable absence of older terraces (Pandey et al., 2014). Further changes in the intensity of the Indian Summer Monsoon during the Holocene have been inferred from marine sediments in the Bay of Bengal and Arabian Sea, and speleothems from Oman and China (Denniston et al., 2000; Goodbred and Kuehl, 2000; Gupta et al., 2005; Clift et al., 2008; Dixit et al., 2014). Limited terrestrial records from the Indian subcontinent (Dixit et al., 2014) suggest a period of intensified Indian Summer Monsoon during the

early Holocene in response to changes in summer insolation forcing, which is consistent with terrace formation driven by enhanced fluvial incision during the early Holocene (Gupta et al., 2005; Srivastava et al., 2008; Sinha et al., 2010; Ray and Srivastava, 2010). Mean sediment flux to the lower Ganga Plains during the period 11-7 ka is estimated to have increased by over two fold (Goodbred and Kuehl, 2000; Sinha and Sarkar, 2009), which is in good agreement with stalagmite $\delta^{18}O$ profiles in Oman which indicate a rapid increase in Indian Summer Monsoon precipitation between ∼10.6 and 9.2 ka (Fleitmann et al.,

2007). Arabian Sea records further indicate an earlier period of monsoon intensification at ∼13 ka, representing the major transition between the glacial and Holocene periods, although smaller magnitude changes in climate are observed even earlier (Sirocko et al., 1993). These phases of incision during the early Holocene are punctuated by minor depositional events that form sequences of fill terraces close to the mountain front. Slip on the underlying Himalayan Frontal Thrust (HFT) produces vertical displacement rates of 4 to 6.9 mm yr$^{-1}$ and may result in terrace abandonment (Sinha et al., 2010). During the mid-Holocene,

stalagmite records in Oman and Yemen suggest that the ISM has been gradually weakening since ∼7.6 ka in response to a progressive decrease in summer insolation (Fleitmann et al., 2007). Evidence presented by Gupta et al. (2005) suggests that the ISM entered a more arid phase at ∼5 ka, although a number of abrupt events punctuate the mid to late Holocene record. For example, speleothem evidence from caves in central Nepal has suggested that between 2300-1500 yr BP there was a significant drop in monsoon precipitation (Denniston et al., 2000; Fleitmann et al., 2007). In general however, the ISM appears to have

been relatively stable over the last 1.5-2 ka.

A number of slack water and flood deposits in the Ganga valley record rapid sediment accumulation over the Ganga flood-plain during high flow events in the late Holocene (Wasson et al., 2013). Seven of these flood units have been dated between ∼280 and 600 years old by OSL and calibrated with $^{14}C$ ages from preserved charcoal fragments (Wasson et al., 2013). These deposits are preserved in a slightly wider part of the bedrock gorge upstream of the mountain front, where flood waters would

have backed up as the river enters the narrower gorge immediately downstream. Additional deposits were studied by Wasson et al. (2013) at Devprayag and Raiwala (Fig. 1) although they recorded small flood couplets as opposed to single flood event deposits. Stacked sand-silt couplets representing phases of persistent flooding were also identified between 2,500-1,200 and 320-209 yr BP at Devprayag and were attributed to changes in the spatial extent of the ISM based on geochemical evidence (Srivastava et al., 2008).



During 2013, heavy rainfall between the 15[th] and 17[th] June was centred over the Alaknanda and Bhagirati catchments and generated significant flash flooding and numerous landslides, causing notable damage to the Kedarnath region in the Alaknanda catchment (Fig. 1). A moraine dammed lake (Chorabari) had formed north west of the Kedarnath region in response to the elevated levels of snow-melt runoff in the preceding month, which is also understood to have burst on the morning of 17[th] June 2013, releasing water with a peak discharge estimated at 783 m$^3$ s$^{-1}$ into the Alaknanda valley (Durga-Rao et al., 2014). Flash flooding is not an uncommon phenomenon in the Ganga basin; other large magnitude events were documented in 1894 and 1970 (Rana et al., 2013). Both of these flood events were attributed to the breaching of dams created by landslides on the tributaries of the Alaknanda River, following unusually high rainfall events. Sediment deposited following the 2013 floods upstream of Devprayag (Fig. 1) over-topped the 1970 flood sediment deposits (thought to be the largest flood during the last 600 years), suggesting that the 2013 flood water levels were the highest in the Alaknanada valley during at least the last 600 years (Rana et al., 2013; Wasson et al., 2013), and possibly since the Last Glacial Maximum (Devrani et al., 2015). The 2013 event also presents a rare opportunity to re-sample $^{10}$Be concentrations following an extreme flood event in the modern Ganga River, to compare against pre-event concentrations as documented by Lupker et al. (2012).

## 3    Methods

Quartz-rich sand samples were taken from modern gravel bars (herein termed modern samples) and independently dated terrace and floodplain deposits (Fig. 3). $^{10}$Be concentrations measured from floodplain samples are thought to accurately reflect upstream basin-averaged denudation rates if sediment residence time in the floodplain is sufficiently short to avoid additional $^{10}$Be accumulation prior to burial (Gosse and Phillips, 2001; Lupker et al., 2012). In the instance of thick event beds (>2 m), sediment at the base of each bed is assumed to have been rapidly buried to a depth greater than the penetration range of cosmic rays, so will have remained shielded since burial and therefore should have accumulated minimal post-depositional $^{10}$Be. In order to reduce the impact of $^{10}$Be accumulation after deposition of dated terraces, sediment samples were collected from the base of thick beds (> 1 m) that record individual flood events either as overbank fines, or as channel braid bars (Wasson et al., 2013). At least 2 kg of quartz-rich sand was sieved from the base of event beds. All samples were collected following horizontal digging for ∼1 m into steep cuts through the deposits to minimise post-burial CRN production. CRN concentrations from terrace and floodplain samples were corrected for post-depositional $^{10}$Be accumulation by considering that the samples had been exposed to cosmic radiation since deposition at the same depth as they were sampled from. For the slower, long-term sedimentation rates of ∼2 mm yr$^{-1}$ in the older early Holocene terraces, only samples from the base of very thick-bedded (>1-2 m) gravels were used to minimise post-depositional effects, where it is assumed that samples would have been largely shielded from further CRN production. Sample depths and post-depositional corrections are presented in Table 1. Sand was taken from the base of several metre thick sand deposits (RFLO and DV2013) abandoned following the summer 2013 Alaknanda flood event to evaluate the degree of mixing of sand during a single extreme event.

Floodplain, terrace and modern river sand samples were first dried before sieving into a number of grain size fractions. The main grain size fraction of interest in this study is 250-500 $\mu$m. Samples with sufficient material in the 250-500 $\mu$m



fraction were then passed through a horizontal Frantz to remove magnetic minerals. Samples were also supplemented with material from the 125-250 $\mu$m grain size fraction where there was insufficient material in the 250-500 $\mu$m fraction. Following this procedure, samples were put through repeated dissolutions in aqua regia and diluted HF and $HNO_3$ solutions to remove mineral phases other than quartz. Quartz samples were then etched with HF to remove between 30 and 50 % of their volume.

The purity of the clean quartz cores were then tested by ICP-OES. All the Al concentrations in the quartz cores were below 300 ppm. Between 7 and 30 g of quartz cores were dissolved in concentrated HF. Samples were spiked with c. 220 $\mu$g of a $^9$Be carrier produced in the cosmogenic isotope analysis facility at the Scottish Universities Environmental Research Centre (SUERC) from phenakite crystals. The $^{10}$Be carrier concentration is c. 9 $\times 10^{-16}$ $^{10}$Be/$^9$Be. A procedural blank was prepared together with each group of samples. Be was isolated from the solutions following routine column chemistry (Darvill et al.,

2015). $^{10}$Be/$^9$Be ratios of the produced BeO targets were measured with the 5 MV Pelletron AMS at the SUERC (Xu et al., 2010). $^{10}$Be data were calibrated against the National Institute of Standards and Technology standard reference material NIST SRM 4325. The activity of NIST SRM 4325 corresponds to a nominal $^{10}$Be/$^9$Be ratio of 2.79 $\times 10^{-11}$ for a $^{10}$Be half-life of 1.36 $\times 10^6$ years. The processed blank ratios ranged between 4 and 54 % of the sample $^{10}$Be/$^9$Be ratios. The uncertainty of this correction is included in the stated standard uncertainties.

Catchment-averaged denudation rates were calculated for each sample using the CAIRN method (Mudd et al., 2016), which estimates production and shielding factors on a pixel-by-pixel basis, rather than a catchment-averaged shielding factor as in more commonly used CRN analysis packages such as CRONUS (Balco et al., 2008). Snow shielding was determined for the Ganga catchment using data downloaded from the Global Land Ice Measurements from Space (GLIMS) Glacier Database (Armstrong et al., 2005); production rates beneath snow covered areas were assumed to be zero. The GLIMS data suggest that

~14 % of the Ganga catchment is glaciated (Fig. 1), which is ~12 % higher than estimates in Lupker et al. (2012) which were produced prior to the completion of the GLIMS database in this region. The proportion of catchment glacier cover is likely to have been notably higher during the early Holocene, and as such, production rates may have been lower when averaged over the full catchment. We therefore consider the production and erosion rates calculated for ancient deposits as maximum values.

## 4   Results

The $^{10}$Be concentrations of the two modern samples near the mountain front (GAPUB and RAEM) are 17.70 and 13.56 $\times 10^3$ at g$^{-1}$, respectively. When combined with sample LUPK09 from (Lupker et al., 2012) which was similarly collected near the mountain front, an average concentration of 14.1 $\times 10^3$ at g$^{-1}$ is estimated for modern samples. $^{10}$Be concentrations of the majority of samples, both from ancient terraces and recent flood deposits, largely fall within the error of modern detrital samples (Fig. 4 and Table 1). Only three samples (BG1.8, DVDF and CDT4) display $^{10}$Be concentrations considerably greater

than the upper error bound (19.1 $\times 10^3$ at g$^{-1}$) of modern river samples; the average concentrations of these terrace samples are in excess of 20 $\times 10^3$ at g$^{-1}$. Only one sample, DVTT2, has an average concentration (6.66 $\times 10^3$ at g$^{-1}$) notably below the lower error bound of the modern samples (8.20 $\times 10^3$ at g$^{-1}$). Samples taken from flood deposits associated with the 2013 Alaknanda



flood (DV2013 and RFLO) reveal concentrations of 16.06 and 12.85 $\times 10^3$ at $g^{-1}$, respectively, which fall well within the error of modern river sediment samples.

In a frequency-histogram of $^{10}$Be concentration data (Fig. 5a), the three samples with the highest concentrations (BG1.8, DVDF and CDT4) produce a positively skewed distribution. These samples represent a fine grained $\sim$300 year flood deposit (Wasson et al., 2013), $\sim$10,000 year old terrace fill (Srivastava et al., 2008) and $\sim$11,000 year old terrace fill (Sinha et al., 2010), respectively (See Appendix B for further sample details). With the removal of samples BG1.8 and CDT4 from the frequency-histogram, the $^{10}$Be concentration data generate a near-normal distribution (Fig. 5a). Possible explanations for the high concentration measurement at BG1.8 may include insufficient shielding since deposition, resulting in $^{10}$Be enrichment of the deposit. Unlike other samples analysed here, the event bed associated with this sample was only $\sim$0.5 m thick so burial (and therefore complete shielding) was unlikely to be instantaneous. Whilst a number of additional samples were taken from this exposure to try and produce depth-concentration profiles, their grain size was too fine for CRN analysis. However, the maximum CRN enrichment at the site during burial is likely to only be $\sim$1650 atoms $g^{-1}$ based on local CRN production rates and sample depth, which is less than the measurement uncertainty. With respect to the two terrace deposits (DVDF and CDT4), high concentrations could also have been produced if the samples were overwhelmed by locally derived, high concentration hillslope sediment which was not well mixed. Samples with the largest CRN concentration variability also seem to focus around 10-15 ka (Fig. 4), which may represent a period of post-glacial conditions where a combination of low CRN concentration material (generated by glacial erosion) and high CRN concentration sediment (due to lower precipitation rates and therefore slower erosion of non-glaciated landscapes) generated during the Last Glacial Maximum may have been mobilised as the ISM intensified during the early Holocene.

Results from CAIRN modelling of all concentrations suggest that catchment-averaged denudation rates for each sample largely lie within the error of modern detrital samples (Fig. 5b). Based on the measured concentrations, these samples correspond to integration timescales of $\sim$500 years, representing the average time period when the erosion rate is considered to be constant, based on the time needed to erode one mean attenuation path length (approximately 60 cm/erosion rate) (Lal, 1991). There does not appear to be a spatial trend between $^{10}$Be concentration and upstream catchment area, even downstream of large tributary confluences (Fig. 6). The impact of high CRN concentration samples on the frequency-histogram of erosion rates calculated using CAIRN modelling is less apparent (Fig. 5b), but the distribution shows significant spread. Calculating sediment flux estimates from a single erosion rate at the upper end of the distribution could result in sediment flux estimate being up to seven times larger than one based on a sample at the lower end of the distribution.

## 5 Impact of stochastic inputs on CRN variability and sediment flux estimates

### 5.1 Impact of landslides on CRN variability

A range of processes are likely to drive temporal variability in CRN concentrations in sand sampled close to the outlet of large Himalayan catchments. The most obvious process is stochastic inputs generated by mass wasting of hillslopes, which generate large quantities of sediment with relatively low CRN concentrations. Frequency-histograms presented in Figure 5 suggest that





such stochastic processes may form part of the natural background variability, as low concentration values tend not to skew the distributions. More samples would be needed to draw sa clearer picture on this. Below, we examine how different erosional processes may drive the observed variability in CRN concentrations measured close to the Ganga outlet. This is approached using a numerical analysis of catchment-averaged CRN concentrations derived under varying background erosion rates, lands-

lide area, depth and surface CRN production rates. Given the complexity of this type of landscape (e.g. multiple geomorphic process domains, climatic variability), we do not attempt to mimic these processes and reproduce measured concentrations. Neither do we use this analysis to determine the relative contributions required from stochastic processes (e.g. area and depth of landsliding) to produce our observed concentrations. Instead, this numerical analysis is used to explore the sensitivity of outlet CRN concentrations to a range of parameters and scenarios that may drive variability.

The relative [10]Be contribution by landsliding can be approximated to first-order by calculating the volume of material generated by the event, and the average concentration of that material. The concentration of landslide material is strongly controlled by the local surface CRN production rate and depth of the landslide. CRN production rates rapidly diminish in the upper few metres of the Earth's surface (Lal, 1991; Stone, 2000; Niedermann, 2002) following:

$$P(z) = P_0 e^{\left(\frac{-z\rho}{\Lambda}\right)} \tag{1}$$

where $z$ is the depth below the surface (cm), $\Lambda$ is the attenuation length (g cm$^{-2}$), $\rho$ is rock density (g cm$^{-3}$), and $P_0$ is the surface nuclide production rate (atoms g$^{-1}$ yr$^{-1}$). At depths greater than $\sim$2 m the CRN production rate (by spallation reactions) is negligible, as is muon production, as atoms generated by muon interactions represents a small proportion relative to those produced by spallation reactions in the upper 1-2 m of the Earth's surface (e.g. Niedermann, 2002). Here, we calculate the average concentration of landslide material by integrating the surface production rate within the upper 2 m; we find that the

depth-averaged production rate of the upper 2 m ($P_d$) is $\sim$30 % of $P_0$. This was converted into a [10]Be concentration ($C$) in atoms g$^{-1}$ using:

$$C = \frac{(P_d \Lambda)}{\rho(\epsilon + \Lambda \lambda / \rho)} \tag{2}$$

from Niedermann (2002), where we assume that the CRN decay constant ($\lambda$) is equal to 0 over the timescales we are concerned with ($<10^3$ years) relative to the half-life of [10]Be. We use $\rho$ = 2.7 g cm$^{-3}$ and $\Lambda$ = 160 g cm$^{-2}$. We also assume a

steady-state erosion-rate ($\epsilon$) across the upstream catchment. For landslide depths of less than 2 m, the average concentration was calculated based on the production rate integral specific to that depth. For simplicity, we assume that the rest of the catchment is eroding uniformly at a background erosion rate, with a catchment average CRN production rate of 35 atoms g$^{-1}$ yr$^{-1}$ which is comparable to the catchment-averaged production rate calculated for the Ganga catchment in CAIRN. The





concentrations calculated at the Ganga outlet also assume complete sediment mixing. The CRN concentration at the catchment outlet ($\alpha_{event+uniform}$) is then calculated using:

$$\alpha_{event+uniform} = \frac{(\alpha_{uniform}\phi_{uniform}) + (\alpha_{event}\phi_{event})}{\alpha_{uniform} + \alpha_{event}} \qquad (3)$$

where $\phi_{uniform}$ and $\alpha_{uniform}$ are the background sediment flux and $^{10}$Be concentration, respectively. $\phi_{event}$ and $\alpha_{event}$ are the

event or landslide generated sediment flux and $^{10}$Be concentration, respectively. A series of sub-catchments were then selected to examine the influence of spatial variability in surface production rates across the Ganga basin, to provide a realistic range of values in the numerical analysis (Fig. 7). Average shielding factors (snow and topographic shielding) were first calculated for each of these sub-catchments using the CAIRN method (Mudd et al., 2016), which were then used in the online CRONUS v2.3 calculator (Balco et al., 2008) to calculate production rates, using a constant production rate model with a Lal/Stone scaling

scheme for spallation (Fig. 7 and Table 2). The default landslide surface production rates were initially set to the same as the catchment-average production rate. The landslide surface production rates were then varied based on realistic production rates derived from sub-catchments across the Ganga catchment (Table 2). Earthquake-induced landsliding datasets from the 1999 Chi-Chi (Taiwan) and 2015 Gorkha (Himalaya) earthquakes (Lin and Tung, 2004; Martha et al., 2017), state that the total landslide areas were ~128 and 90 km$^2$, respectively. Areas of these sizes represent approximately 0.5 % of the Ganga

catchment area. We therefore use the value of 0.5 % as an approximation of the proportion of the hypothetical catchment to have been impacted by landsliding. In the analysis, the average depth of the landslides was varied from 0.5 to 5 m, the average background erosion rate from 0.2 to 2.0 mm yr$^{-1}$, and the average landslide surface production rate from 10 to 60 atoms g$^{-1}$ yr$^{-1}$. We use an average landslide depth where in reality, the depths of individual landslides occurring in response to an earthquake or intense storm are likely to fit a power-law distribution (Hovius et al., 1997). However, at any point in time it is unlikely that the

full power-law distribution of landslide depths is sampled or integrated into the catchment wide signal, due to the recurrence interval and amount of time taken to evacuate larger and deeper landslides. We also assume that the CRN concentration profile in the upper 2 m of the landscape is in steady-state before landsliding. This assumption is more important in slowly eroding landscapes, where it may take tens of thousands of years to reach secular equilibrium (Dunai, 2010). This may result in over-estimated landslide CRN concentrations in our analysis, if the CRN concentration profile is not in equilibrium. However, by

varying the landslide surface production rates in our analysis we can indirectly assess the importance of such an effect.

We calculate 'volumetric sediment flux' by combining the flux derived from background erosion rates with the calculated landslide flux, and compared these to sediment flux estimates derived from the $^{10}$Be concentration at the catchment outlet (which we term the 'CRN-derived sediment flux'). For a catchment eroding at a uniform rate ($\epsilon$ in mm yr$^{-1}$), the CRN-derived sediment flux is the product of the erosion rate, catchment area ($A$ in km$^2$) and average rock density ($\rho$ in kg m$^{-3}$).

In this analysis, we assume that sediment storage between the region affected by landslides and the outlet is small relative to the total sediment flux of the catchment. Unlike the eastern and western Himalaya, the central Himalaya (which is largely drained by tributaries of the Ganga River) is comparatively void of large valley fills (Blöthe and Korup, 2013), which is likely to limit large volumes of sediment storage and sediment residence times. Recent modelling has also suggested that approximately





50 % of coarse material generated by post-seismic landsliding is evacuated within 5 to 25 years (Croissant et al., 2017). In our scenarios, we assume complete evacuation of material to the outlet within a year. The effect of reducing the amount of landslide derived sediment contributing to the outlet concentration are shown in the Supplementary Material for reference however. The default and range of values tested for each parameter in the analysis are shown in Table 3.

Based on the above calculations, our results suggest that increasing the average landslide depth results in a marked decrease in outlet $^{10}$Be concentration, most notably between depths of 0.5-3 m (Fig. 8a). This can be explained through the exponential decay in $^{10}$Be production rates in the upper 2 m of the landslide (Lal, 1991; Stone, 2000; Niedermann, 2002). This reduction in concentration is greatest under lower background erosion rates. Increasing background erosion rates from 0.2-2.0 mm yr$^{-1}$ also reduces the effect of landsliding on outlet $^{10}$Be concentrations (Fig.3b). Under lower background erosion rate, landslide

material represents a greater proportion of the total sediment flux, so the system has less capacity to buffer the landslide input and the $^{10}$Be concentration is more sensitive to deeper landslides. We also find that outlet $^{10}$Be concentrations are sensitive to the average landslide surface production rate. Where the average surface production rate of the landsliding is increased (e.g. comparable to that expected in high altitude sub-catchments of the Ganga - see Table 2), predicted outlet $^{10}$Be concentrations also increase relative to scenarios with otherwise identical parameter values (Fig. 8c). Interestingly, we also find that volume-

tric sediment flux estimates are consistently higher than CRN-derived fluxes (Fig. 8d). Increasing background erosion rates increases both CRN-derived and volumetric sediment flux estimates, but increasing average landslide depth or landslide CRN production rate can reduce CRN-derived sediment flux estimates to a much greater degree than volumetric flux estimates.

Our analysis generates variability in CRN concentrations that is considerably larger than what we document in the Ganga catchment (Fig. 4), suggesting that buffering of stochastic inputs must occur (Croissant et al., 2017). The evacuation time of

fine-grained sediment (sand and finer) is likely to be fast relative to the coarse fraction, as the fine-grained fraction is annually entrained and transported downstream during months impacted by the Indian Summer Monsoon. This is supported by grain size analysis (Dingle et al., 2016) along a number of exposed gravel bars within the Ganga catchment, which demonstrate that the channel bed is comprised largely of grain sizes >1 mm, even beneath the surface armour layer. Typically, grain sizes <1 mm represent less than ~15 % of the grain size distribution (Fig. 9) which is also observed across other catchments of

the Ganga River. This suggests that there is relatively little in-channel storage (or mixing) of finer grained sediments relative to the large fluxes of these river systems, which on entering the Ganga Plain, are thought to be largely dominated (>90 %) by sand-sized (and finer) sediments (Dingle et al., 2017). However, the majority of landslide deposits are likely to be made of coarser material (Attal and Lavé, 2006; Attal et al., 2015) which will take longer to be evacuated or abraded into smaller and more easily transportable grain sizes. Whilst landsliding may generate the quantities and $^{10}$Be concentrations of sediment

required to drive significant changes in concentration at the outlet, the evacuation timescales of these event sediments buffers their impact. Evacuation of event deposits over decadal to centennial timescales will reduce the ratio of background to event sediment fluxes (Croissant et al., 2017), and likely limit the impact on $^{10}$Be concentrations documented at the outlet.



## 5.2 Other potential sources of variability in CRN concentration

Whilst landsliding with different depths and from different parts of the Ganga catchment is likely to represent a key component in CRN variability, a number of other factors may also contribute, which are discussed below. Firstly, spatially variable distributions of quartz-rich lithologies across the Ganga catchment may lead to over and under-estimation of denudation rates in specific lithological settings. However, potential variations in sediment quartz content have been assessed by Vance et al. (2003) in the Ganga catchment, who concluded that the correction due to the dilution of quartz from sediments sourced from carbonate-rich series in the catchment is of a similar magnitude (maximum of ∼9 % change in erosion rate for sub-catchments in the High Himalaya) to the production rate estimates and analytical errors. Recent studies have also highlighted the effect of grain-size dependent $^{10}$Be enrichment, where coarser gravel-sized fractions have been documented to yield higher apparent denudation rates than the medium sand-sized fraction which is typically sampled (Puchol et al., 2014; Schildgen et al., 2016; Lukens et al., 2016) as a result of the process through which the different grain size fractions are generated (e.g. reworked hillslope material, landsliding), or differing sediment source elevations. Similarly, downstream lags in $^{10}$Be denudation rate spikes have been observed along the Tsangpo-Brahmaputra River in the eastern Himalayan syntax (Lupker et al., 2017), due to the distance which sediment generated in the rapidly uplifting Namche Barwa-Gyala Peri massif must travel before being abraded into the grain size fraction used for sampling. However, modern samples collected close to the Ganga outlet are not likely to be influenced by either process, as the majority of sediment has already been abraded into sand by this point (Dingle et al., 2017). Similarly, a number of the floodplain and terrace deposits sampled were entirely sand. Exceptions to this include terrace deposits CDT3, CDT4, DVDF, DVMT2, DVTT2 and RLB, where sand samples were taken from poorly consolidated fluvial deposits containing imbricated and well-rounded quartzite cobbles and pebbles. However, additional CRN samples were not run on individual clasts in these deposits to determine whether the coarser fraction yielded higher apparent denudation rates.

Glacial lake outburst floods (GLOFs) are not uncommon across the Himalaya (e.g. Cenderelli and Wohl, 2003; Kattelmann, 2003), and have the potential to generate and mobilise large quantities of sediment. Geomorphic analysis following the 1977 and 1985 GLOFs in the Mount Everest region (Cenderelli and Wohl, 2003) suggested that much of the sediment eroded from the upper 10-16 km of the GLOF route was unconsolidated sediment (glacial till, colluvium, glacio-fluvial terraces). Erosion was typically found to be limited in valleys with resistant bedrock or consolidated side walls. Similarly, the availability of unconsolidated material is also thought to be a key limiting factor in the volume of debris flows triggered following GLOFs, which can limit the erosive potential of the flow (Breien et al., 2008). In the absence of existing studies which document $^{10}$Be concentrations in proglacial lake sediments, we cannot infer how sediment released from the glacial lake may contribute to downstream variations in $^{10}$Be concentration. Geomorphological evidence in reaches downstream of GLOFs suggests that much of the sediment eroded by the flood is largely unconsolidated (glacially-influenced) material from relatively shallow depths (<3 m; Cenderelli and Wohl, 2003) which is likely to have a complex exposure history. Given the relatively short length of the reach impacted downstream of the GLOF (relative to the full length of a system such as the Ganga), and the likely CRN enriched nature of surface deposits reworked by GLOFs, it seems unlikely that these types of events drive significant change in outlet $^{10}$Be concentrations. This is supported by work in the Marsyandi River catchment in Nepal, which suggested that erosion





in the upper glaciated catchment is almost an order of magnitude lower than fluvial incision rates in the upper Marsyandi River (Heimsath and McGlynn, 2008).

Extreme monsoonal storms, such as the one that generated the 2013 Alaknanda flooding, also have the potential to generate CRN variability if hillslope runoff mobilises large quantities of unconsolidated sediment on valley sides and initiates mass-

wasting of hillslopes (Dobhal et al., 2013; Devrani et al., 2015). Sample DV2013 was collected from a thick sand unit at the Ganga channel margins (∼18 m above the modern channel) near Devprayag, known locally to have been deposited following the 2013 Alaknanda flood. We find that the $^{10}$Be concentration of this deposit (16.06 $\times 10^3$ at g$^{-1}$) also lies within the error of modern samples at the outlet. One interpretation is that the sediment generated by this event was sufficiently well mixed: on reaching the Ganga outlet it had minimal impact on the outlet CRN concentration. Material mobilised by the Alaknanda

flooding was largely unconsolidated, surficial hillslope material (Dobhal et al., 2013). As such, the $^{10}$Be concentration of these sediments will reflect their local production rate (∼50 atoms g$^{-1}$ yr$^{-1}$ - see Table 2) and background erosion rate. If erosion in the Alaknanda valley is driven primarily by large storm and flood events, unconsolidated surface sediments could have been accumulating $^{10}$Be since as early as the LGM (Devrani et al., 2015), with very low background erosion rates. As such, this type of erosive event may have generated sediment with a higher than expected CRN concentration (given the depth of material

removed) as a result of this CRN-enriched surface layer.

Annual monsoonal storms may also contribute to the observed variability where storms tap into localised parts of the catchment. The hillslope sediments and reworked deposits these storms mobilise could vary in $^{10}$Be concentration in the different geomorphic process domains, as they will have variable CRN production rates (which is a function of elevation), background erosion rates and deposit characteristics (e.g. deep-seated landslide). Background erosion rates in particular are likely to vary

dramatically across the Ganga catchment as a result of spatially variable rock uplift, lithology, rainfall and vegetation cover (Vance et al., 2003; Anders et al., 2006; Bookhagen and Burbank, 2006). Earthquake-induced landsliding, GLOFs and extreme storm events are all likely to generate large quantities of sediment with $^{10}$Be concentrations that would be sufficient to drive significant change in the $^{10}$Be concentration recorded at the Ganga outlet. However, the impact that these processes have is limited by the ability of the river to entrain and transport this sediment out of the catchment. The evacuation timescales of

sediment generated by these processes will likely vary as a function of the frequency and magnitude of localised storm events which mobilise mass-flow deposits from hillslopes into rivers sediment.

If this sediment is sourced close to the sampling location, it is also unlikely to be fully homogenised. The distance required to fully mix localised hillslope or tributary inputs has been shown to be as much as several kilometres (Binnie et al., 2006), which may induce variability in $^{10}$Be concentrations recorded at the outlet. In terms of modern river samples, a number of small

ephemeral streams drain directly in the main Ganga channel near the outlet. During the monsoon season when these channels are active, sediment of differing $^{10}$Be concentrations will be transported to the main channel and may not be sufficiently mixed on reaching the outlet sampling locations. High concentration samples documented close to the Ganga outlet could therefore represent locally derived and poorly mixed sediments, which reflect the erosional processes specific to a small frontal region of the catchment.





### 5.3 Suitability of CRN as a proxy for sediment flux in large catchments

To understand the observed doubling in sediment delivery to the Bengal fan during the early Holocene, we explore whether it is possible to increase volumetric sediment flux whilst maintaining $^{10}$Be concentrations within the natural degree of background variability. We use the same numerical analysis and run two scenarios. The first scenario represents a baseline condition; in the

second scenario, we increase the background erosion rate, average landslide depth, landslide surface production rate and area of landsliding (Table 4). Landslide mapping following the 2015 Gorkha and Dolakha earthquakes suggests an area of $\sim$90 km$^2$ was affected, generating $\sim$0.62 km$^3$ of landslide material based on area-volume scaling (Martha et al., 2017). This corresponds to an average landslide depth of $\sim$6.8 m, indicating our assumed depths (of 2 m) are conservative. As discussed earlier however, it is unlikely that the full-distribution of landslide size or depth would be sampled or integrated into the catchment-wide signal,

and as such, a lower average depth might be more representative. We have varied parameters between the two scenarios based on what might be expected during periods of increased monsoon intensity: greater background erosion rate and landslide frequency. However, we have no constraint on the magnitude of change in these parameters and, as such, we use the numerical analysis in an informative sense to examine how increased storm-induced mass-wasting and background erosion rates might have impacted CRN concentrations in the Ganga catchment. We also assume that only 7 % and 10 % of the material generated

by landsliding is transported to the outlet in scenarios 1 and 2, respectively, based on the 5-25 years timescales of evacuation suggested by Croissant et al. (2017). A marginally higher proportion is used in the second scenario to represent the effect of elevated water discharge as a result of increased monsoonal precipitation.

Using the parameters shown in Table 4, we find that in both scenarios the $^{10}$Be concentration at the outlet is within the magnitude of natural variability documented at the Ganga outlet. Sediment flux estimates generated from $^{10}$Be concentrations

(termed CRN-derived sediment flux) are within natural system variability as they only differ by $\sim$30 % between scenarios (Table 4). Interestingly under both sets of conditions, the volumetric sediment flux (calculated from the volume of material eroded under background erosion across the entire catchment and the imposed landslide dimensions) is higher than the CRN-derived sediment flux. Furthermore, the volumetric sediment flux generated in the second scenario is double the corresponding CRN-derived flux. This is consistent with results from the numerical analysis, where the volumetric sediment flux is typically

at least two to three times greater than the CRN-derived flux (Fig. 8d).

Our results suggest that, for $^{10}$Be concentrations within a natural degree of system variability, the volumetric sediment flux could theoretically differ from that calculated directly from $^{10}$Be concentrations (Fig. 8d and Table 3). Similar outlet CRN concentrations can be derived from landscapes dominated by different erosional processes within large catchments. For example, our analysis suggests that a 'fast eroding' landscape experiencing a background erosion rate of 2.0 mm yr$^{-1}$ and 1

30   m deep landslides over 0.5 % of the catchment (e.g. a landscape dominated by shallow landsliding or debris flows) could produce comparable outlet CRN concentrations to a 'slow eroding' landscape experiencing 0.4 mm yr$^{-1}$ background erosion and 5.0 m deep landslides over the same area (e.g. a landscape experiencing deep earthflows) (Fig. 10). The CRN-derived sediment fluxes between these two landscapes may be comparable, but the volumetric flux from the landscape with lower background erosion (and deeper landsliding) is considerably larger than from the landscape with higher background erosion





(and shallower landsliding). Halving the area affected by landsliding in only the lower background erosion scenario (with deeper landsliding) still yields comparable CRN-derived fluxes (within 15 % of each other, rather than 6 %), but the volumetric flux is double that generated under higher background erosion rates (with shallower landsliding over a larger area) . These types 'slow eroding' landscapes which experience episodes of mass wasting are exemplified by arid parts of the northwest Himalaya,

which generally only experience high intensity rainstorms during abnormal monsoon years where the ISM can penetrate north of the orographic barrier formed by the Higher Himalaya (Bookhagen et al., 2005) (Fig. 2). Similarly, slow moving earthflows in parts of the Eel River catchment in California which is characterised by long and low-gradient hillslopes mobilise huge quantities of sediment which contribute to the majority of the suspended sediment flux from the catchment (Mackey and Roering, 2011). The two end-member models presented in Figure 10 suggest that under different geomorphic process domains,

comparable mean CRN concentrations can be produced through very different CRN concentration populations.

  CRN-derived sediment fluxes are based on an average landscape lowering rate, and thus fail to incorporate the effects of spatially limited deeper inputs of sediment which are characterised by much lower CRN concentrations. Lower rates of background erosion means that sediment eroded off the surface is enriched in CRN (as sediment residence times in the upper 1-2 m of the Earth's surface are longer as a function of lower background erosion rates). This effectively averages out the

15 influence of lower concentration input from deeper inputs, and results in near identical CRN concentrations at the mountain front to a system undergoing only a slightly faster (or more uniform) rate of background erosion. Thus, considerably different volumetric fluxes can be obtained for the same CRN concentration. This may explain the absence of a [10]Be concentration signature of Holocene climate change.

## 6 Conclusions

We present CRN analysis from a variety of modern and Holocene sedimentary deposits in a large trans-Himalayan catchment spanning more than 7000 m in relief, where sediment production is heavily influenced by stochastic inputs. We find a natural degree of variability in [10]Be concentrations documented in the modern channel and Holocene flood deposits preserved near the catchment outlet. These concentrations appear insensitive to regional intensification of the Indian Summer Monsoon, thought to have occurred ∼11-7 ka. We demonstrate that it is possible to generate relatively constant [10]Be concentrations at large

catchment outlets despite significant variability in volumetric sediment flux as a result of spatially variable background erosion rates and erosional processes such as earthquake-induced landsliding and storm events. We suggest that the observed variability is driven by 1) the nature of the stochastic inputs of sediment (e.g. the type of hillslope process, surface CRN production rates, degree of mixing), and 2) the evacuation timescales of these sediment deposits. Sediment deposits generated by processes such as earthquake-induced landsliding, GLOFs or storm events, are typically large in volume and low in [10]Be concentration,

but the time taken to mobilise this sediment out of the catchment limits its impact on catchment-averaged concentrations. We suggest that in landscapes characterised by high topographic relief, spatially variable climate and multiple geomorphic process domains, the use of [10]Be concentrations to generate sediment flux estimates may not be truly representative, as comparable mean catchment CRN concentrations can be derived through dramatically different erosional processes. For a given CRN



concentration, volumetric sediment flux estimates may vary considerably and under certain conditions, CRN concentrations may under-estimate actual erosion rates and hence sediment flux.

*Code and data availability.* The CAIRN software used to calculate erosion rates is available at the LSDTopoTools Github wesite ($http : //github.com/LSDtopotools$) with accompanying documentation ($http : //lsdtopotools.github.io/LSDTT\_book/$). The DEM used

in this analysis (Shuttle Radar Topography Mission 30 m resolution) is freely available from the United States Geological Survey digital globe website ($http : //earthexplorer.usgs.gov/$). Full CRN sample details are provided in Table 1 and text within the manuscript. The equations and parameter values used in the numerical analysis are available in the manuscript text.

*Author contributions.* E.H.D., H.S., M.A. and V.S. collected the samples used in the cosmogenic radionuclide analysis, which A.R. and E.H.D. prepared for analysis at SUERC. E.H.D. designed and carried out the numerical analysis. E.H.D. produced the figures and wrote the

manuscript with discussions and contributions from H.D.S., M.A., and A.R.

*Competing interests.* The authors declare that they have no conflict of interest.

*Acknowledgements.* Elizabeth Dingle is funded under a NERC PhD Studentship (NE/L501566/1) and CRN analysis was undertaken at the SUERC CIAF (under grant application 9150.1014). We would like to thank the International Association of Sedimentologists, British Society of Geomorphology and the Edinburgh University Club of Toronto for their financial support of the fieldwork, and Konark Maheswari for his

assistance in the field. We are also grateful to Maarten Lupker, Shasta Marerro and Simon Mudd for discussions that have helped shape this manuscript.



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



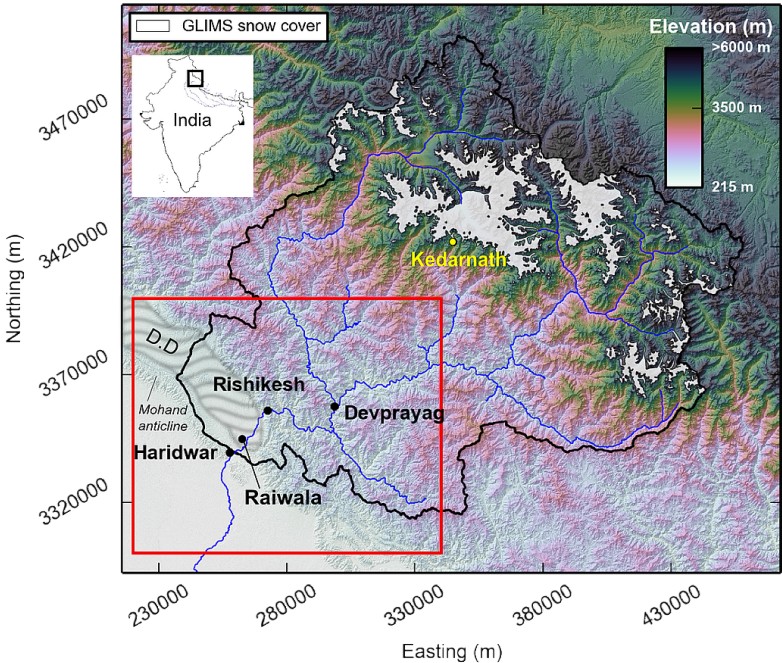

**Figure 1.** 30m Shuttle Radar Topography Mission (SRTM) Digital Elevation Model (DEM) of the Ganga catchment. Coordinates are projected in UTM Zone 44N. Glacier coverage as documented in the Global Land Ice Measurements from Space (GLIMS) database is also shown in white. The red box represents the spatial area shown in more detail in Fig. 3. D.D refers to the Dehra Dun region which is delineated by the grey striped area.

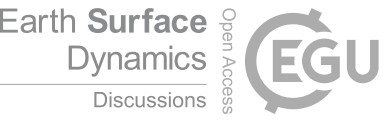

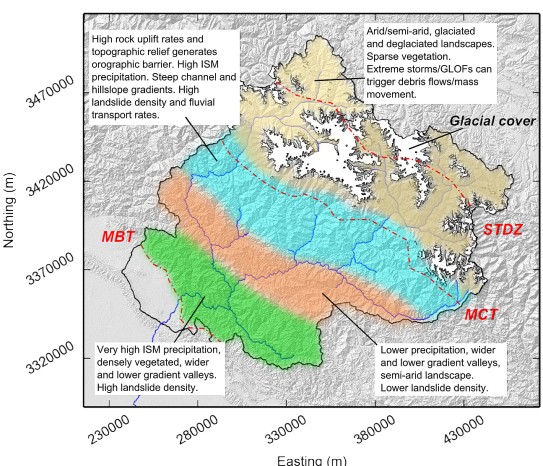

**Figure 2.** Broad distribution of geomorphic process domains across the Ganga catchment. The approximate positions of the Main Boundary Thrust (MBT), Main Central Thrust (MCT) and South Tibetan Detachment Zone (STDZ) are shown by red dashed lines following Ray and Srivastava (2010). Relative landslide density was determined by manual mapping of >400 landslides across the Ganga catchment using GoogleEarth imagery, where landslides in glacially influenced parts of the catchment were excluded. ISM denotes the Indian Summer Monsoon.



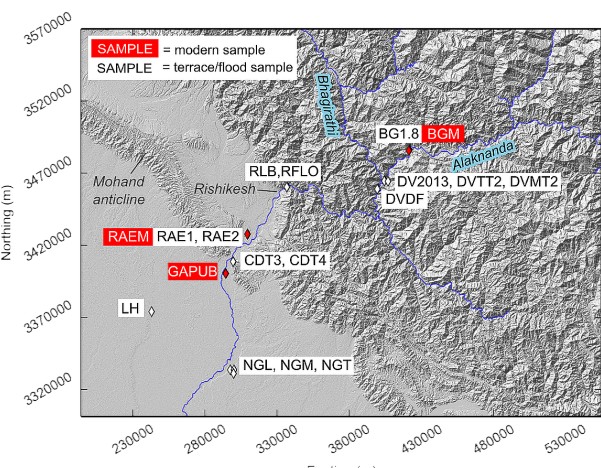

**Figure 3.** Modern (red) and terrace/floodplain/flood (white) sample locations and names in the lower Ganga catchment. See Table 1 for full description of samples.




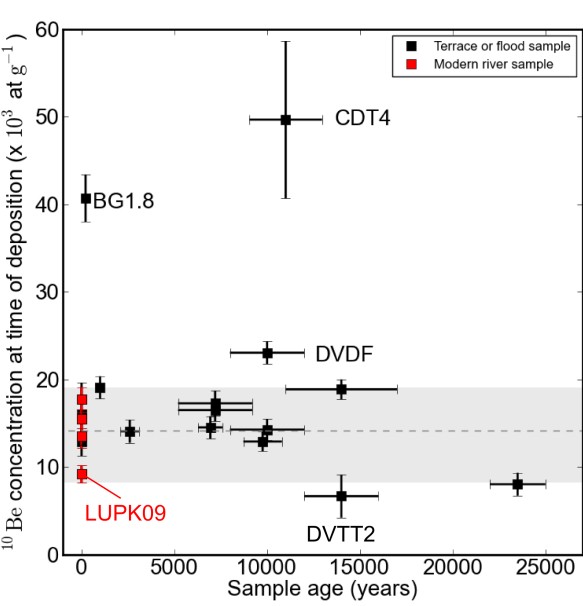

**Figure 4.** Measured modern river (red) and terrace or flood/floodplain (black) $^{10}$Be concentrations relative to their depositional age. Horizontal error bars represent the published age error associated with the independently dated deposit, and vertical error bars represent error in $^{10}$Be concentrations determined in this study. Sample LUPK09 from Lupker et al. (2012) is also included and labelled.



Earth **Surface**
**Dynamics**
Discussions


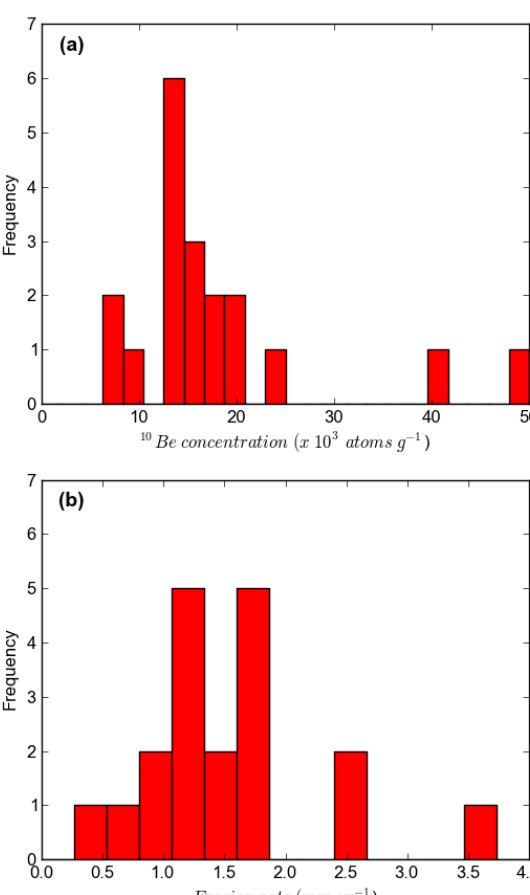

**Figure 5.** (a) Frequency histogram of mean [10]Be concentrations shown in Fig. 4. (b) Frequency histogram of mean erosion rates calculated using the CAIRN method.





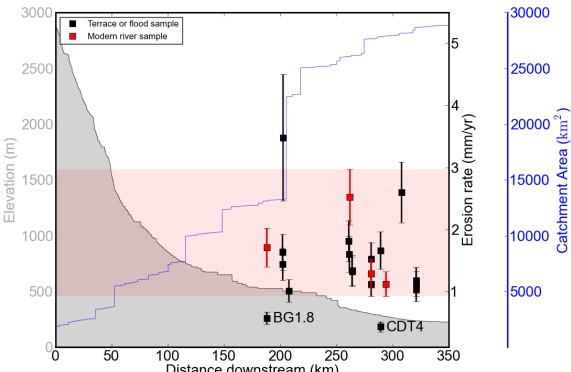

**Figure 6.** Modern river (red) and terrace or flood/floodplain (black) catchment-averaged erosion rates with respect to distance downstream, sample elevation (grey shaded region) and upstream catchment area (blue line). Vertical error bars represent error associated with the modelled erosion rate and propagated $^{10}$Be concentration errors used to derive the erosion rate. The red shaded area represents erosion rates within the error of modern samples. Outliers BG1.8 and CDT4 are labelled.



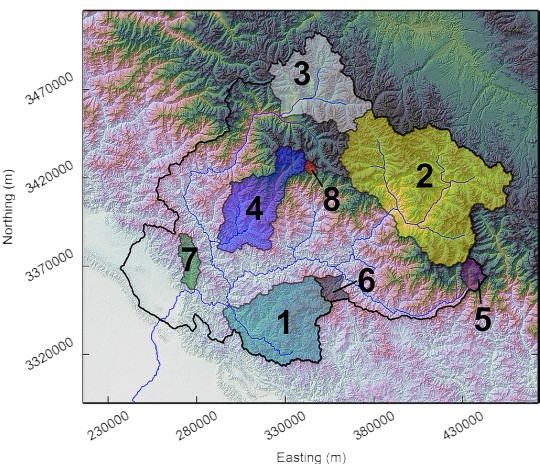

**Figure 7.** Location of sub-catchments used to determine the variability in production rate across the Ganga catchment (presented in Table 2).



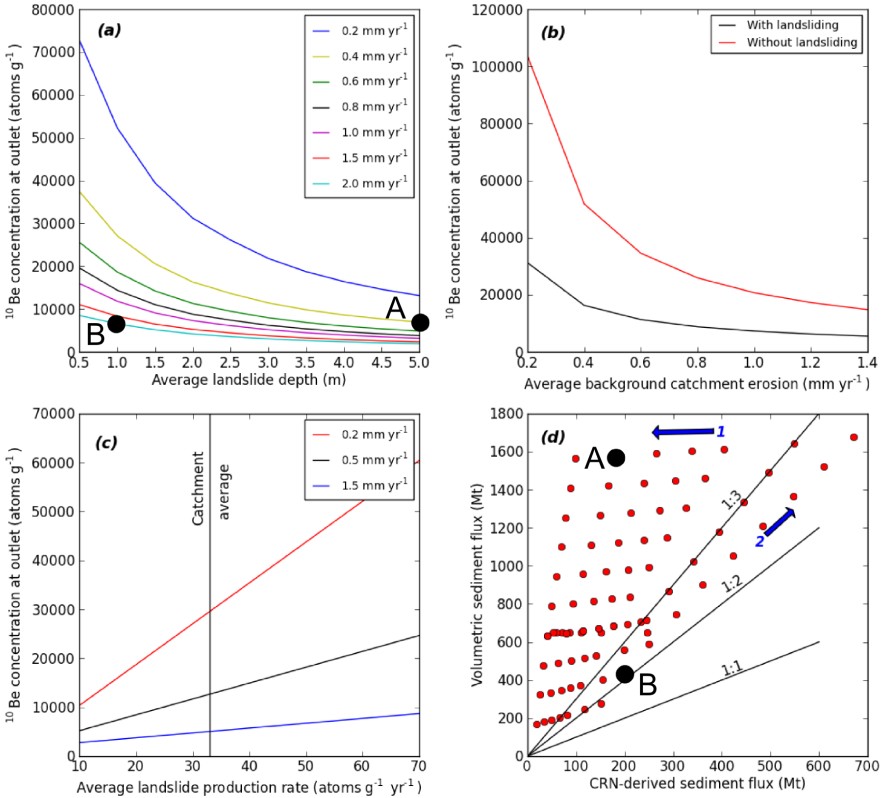

**Figure 8.** (a) Variations in $^{10}$Be concentration predicted at the outlet in response to increasing landslide depth and as a function of background erosion rates (represented by coloured lines). (b) Outlet $^{10}$Be concentration as a function of background erosion rate (where all other parameters are constant at default values - see Table 3), for a system undergoing no landsliding (red line - where erosion is driven purely by background erosion) and another with 2 m deep landsliding over 0.5 % of the catchment area (black line). (c) Outlet $^{10}$Be concentration under varying average landslide $^{10}$Be surface production rates (based on Table 2) and background erosion rates (coloured lines). The black vertical line represents the whole Ganga catchment-averaged production rate of ∼33 atoms g$^{-1}$ yr$^{-1}$. (d) Comparison of volumetric and CRN-derived sediment fluxes from analysis in Figures 8a-c. The blue arrow labelled 1 shows the effect of decreasing background erosion rate, and the blue arrow labelled 2 shows the effect of increasing landslide depth and/or landslide CRN production rate . The black dots in (a) and (d) represent scenarios A and B which are discussed in more detail later and in Fig. 10.





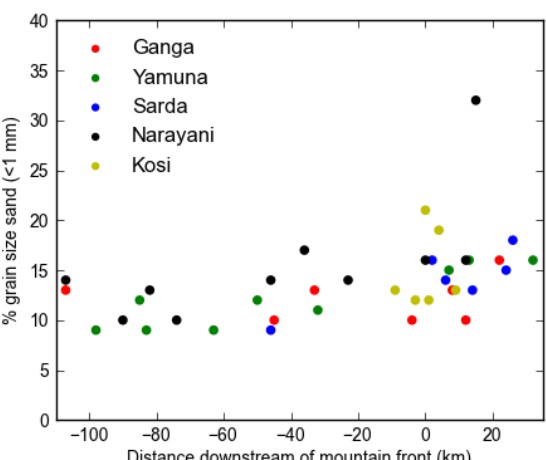

**Figure 9.** Volumetric sand (grain sizes <1 mm) proportions in sub-surface sediment samples along major tributaries of the Ganga River.



Earth **Surface**
**Dynamics**
Discussions



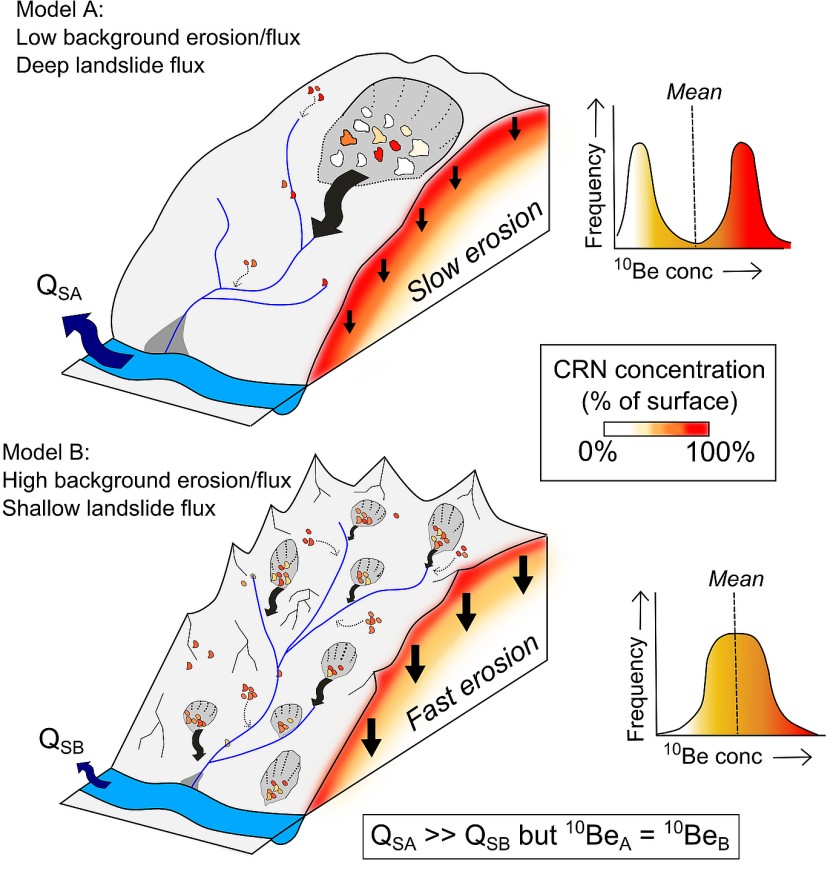

**Figure 10.** Schematic of how comparable mean CRN concentrations in river sand can be derived under two different erosion scenarios with different volumetric sediment fluxes. In these instances, slow background erosion rates and deep landsliding (Model A) result in comparable CRN concentrations to landscapes domianted by faster background erosion rates and shallow landsliding (Model B). If Model A is set with a background erosion rate of 0.4 mm yr$^{-1}$ and 5 m deep landsliding over 0.5 % of the catchment, and Model B with 2 mm yr$^{-1}$ background erosion rates and 1 m deep landsliding (over the same area), comparable CRN concentrations (see black dots marked on Fig. 8a) and CRN-derived sediment fluxes are generated, but volumetric sediment fluxes are over three times larger in Model A. This is due to the relative enrichment of $^{10}$Be in the upper 2 m of the landscape with low background erosion rates, which when combined with low CRN concentration material from depth, results in two distinct CRN concentration populations. Where erosion is generally more homogeneous (Model B) and CRN concentrations are distributed more uniformly, comparable mean CRN concentrations are derived between the two models.



**Appendix A**

The details and context of cosmogenic radionuclide samples used in this study are presented in Fig. A1 - Fig. A16. Locations can also be found in more detail in Fig. 3.



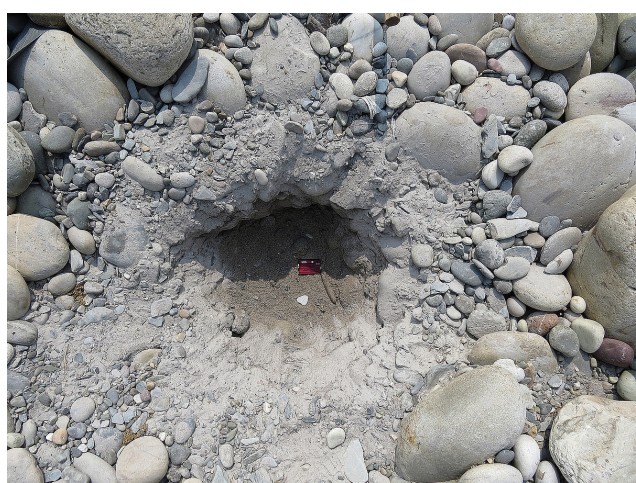

**Figure A1.** BGM - Sieved from upper layer of modern gravel bar. 82 mm long penknife in base of pit.





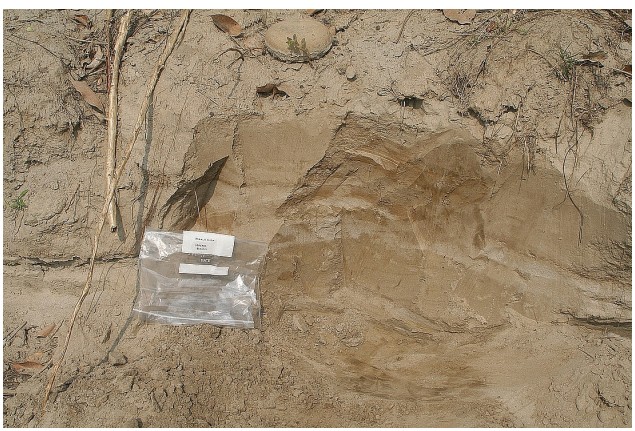

**Figure A2.** BG1.8 - Fine-grained sand deposit (∼7 m in thickness) corresponding to sequence of palaeoflood deposits from last ∼600 years. Sample taken 1.8 m from base of exposure which has been OSL dated at 225±72 years Wasson et al. (2013).





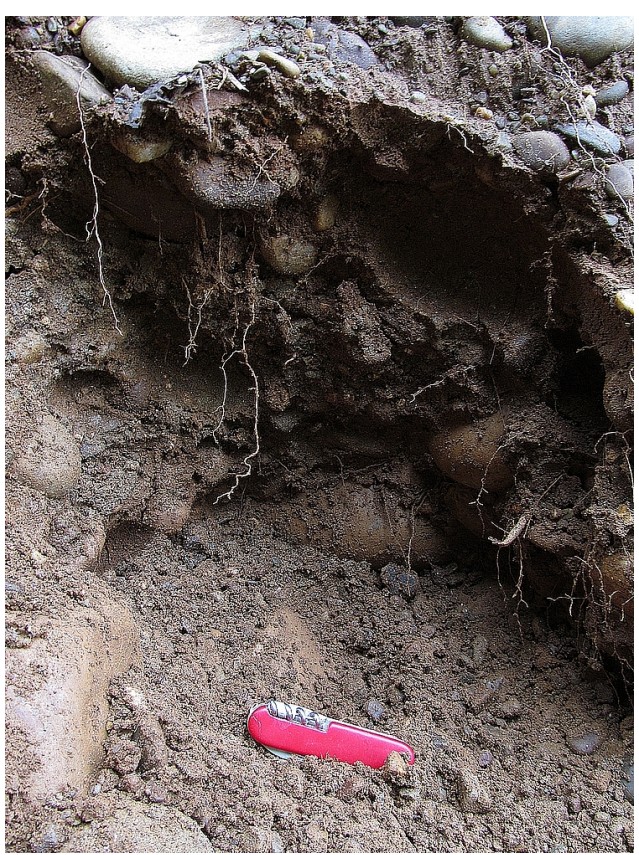

**Figure A3.** CDT3 - Sample from base of ∼3.2 m thick fill of poorly sorted fluvial pebble and cobble conglomerate, suggesting it was deposited during a single event. Approximately 26 m above the modern channel. OSL dated at 9,760 ±1,040 years (Ray and Srivastava, 2010). 90 mm long penknife for scale.



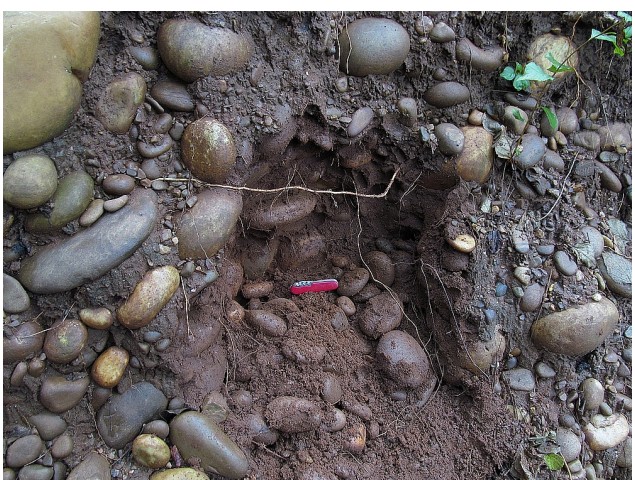

**Figure A4.** CDT4 - Sample from poorly sorted fluvial pebble and cobble conglomerate terrace fill deposited during a single event. Sample ∼3 m below terrace surface and ∼80 m above modern channel. OSL dated at 11,080 ±1,960 years (Ray and Srivastava, 2010). 90 mm long penknife for scale.





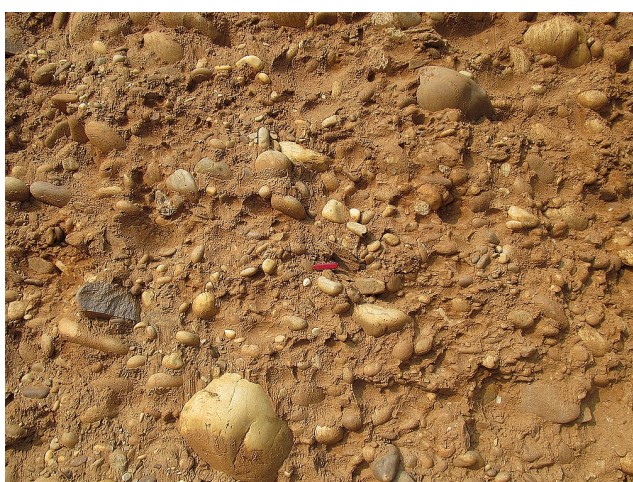

**Figure A5.** DVDF - Terrace deposit ∼95 m above modern channel. Sample taken from base of 4 m thick fluvial conglomerate layer. Capped by more angular phylite/schist deposit (erosional contact) suggesting input of locally derived landslide/debris flow material. Unit OSL dated at 10,000±2,000 years (Ray and Srivastava, 2010). 90 mm long penknife for scale.




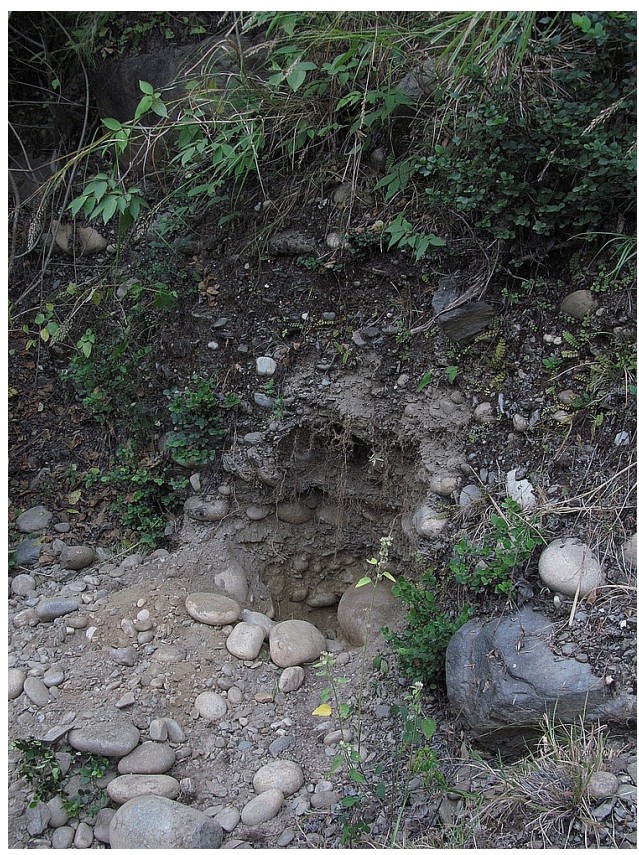

**Figure A6.** DVMT2 - Terrace deposit ∼77 m above modern channel. Poorly sorted and weakly consolidated fluvial pebble and cobble conglomerate. Sample taken from base of 6.5 m unit. Unit OSL dated at 10,000±2,000 years (Ray and Srivastava, 2010).





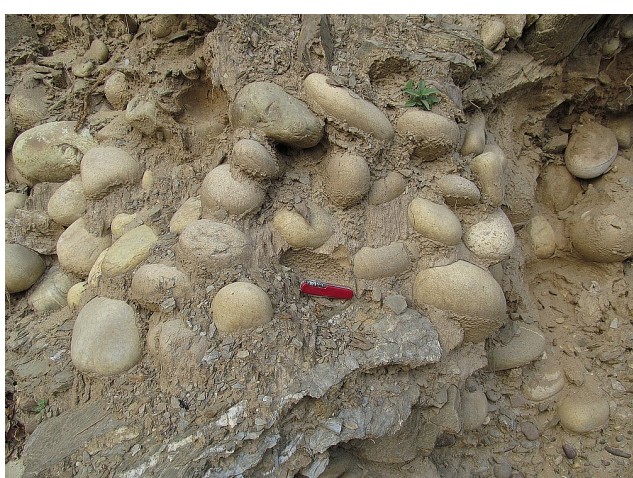

**Figure A7.** DVTT2 - Terrace deposit ∼112 m above modern channel. Fluvially derived coarse cobble and sand (poorly sorted) conglomerate interbedded within locally derived (Lesser Himalayan) phyllite deposits. 90 mm long penknife for scale.




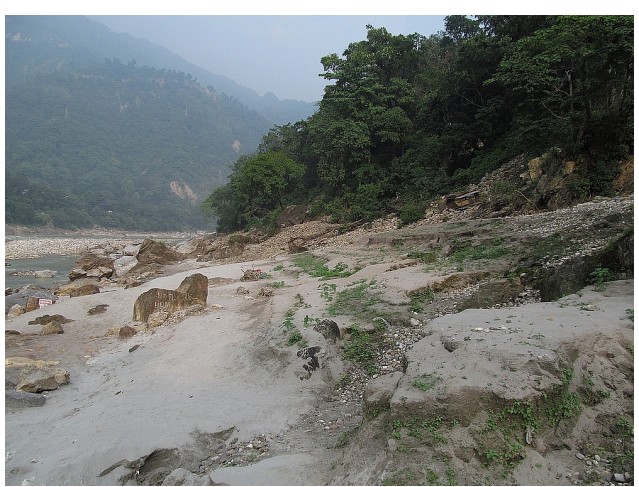

**Figure A8.** RFLO - Sand flood deposit associated with 2013 Alaknanda flooding. ∼7 m above water level in October 2014.



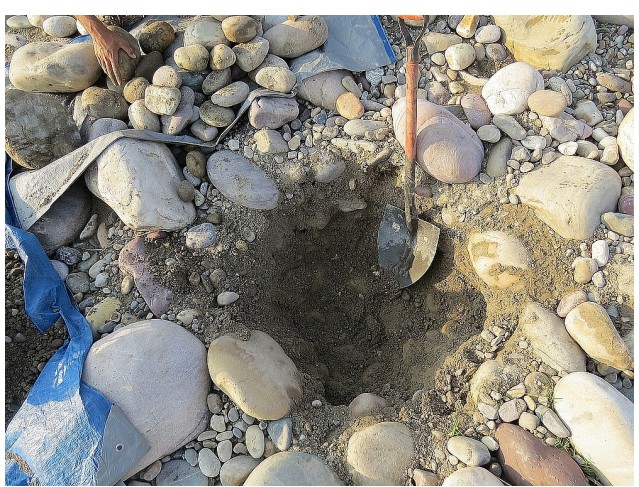

**Figure A9.** RAEM - Sieved from upper layer of modern gravel bar.


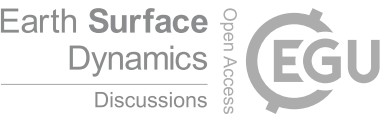

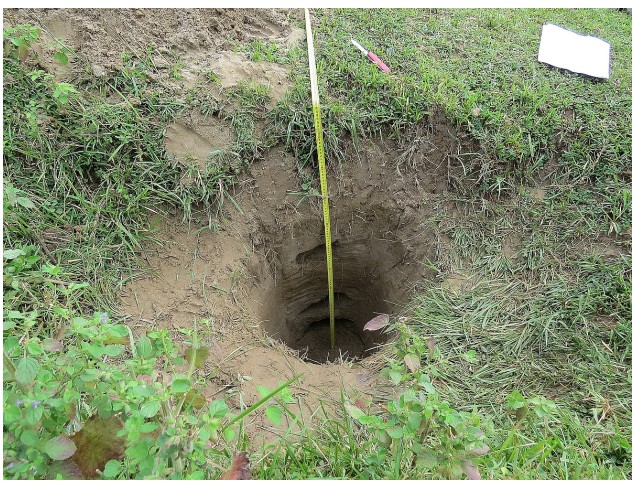

**Figure A10.** RAE1/RAE2 - ∼0.8 m thick sand and silt deposit above cobble bed. Capped by ∼30-50 cm of soil. Samples taken from the lower-most and middle units identified in P1 in Wasson et al. (2013) which are dated at 2.6±0.6 ka and 1.0±0.2 ka, respectively.




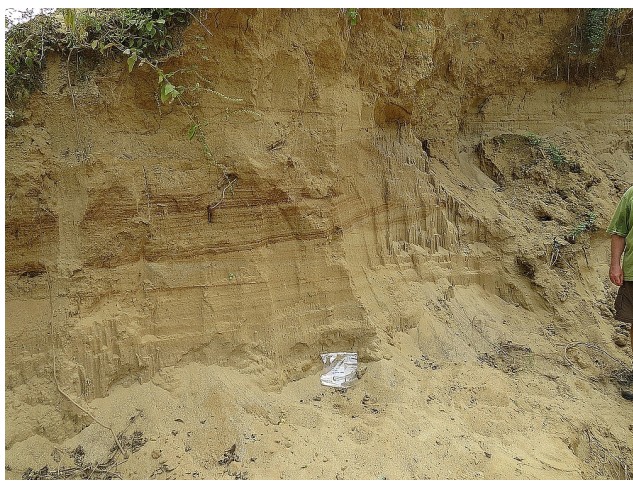

**Figure A11.** NGM - Cross-bedded sand succession ∼17 m above modern channel. Sample taken from base of 1.5 m thick cross-bedded sand unit. Top of unit (S2) OSL dated at 7,200±2,000 years by Sinha et al. (2010).



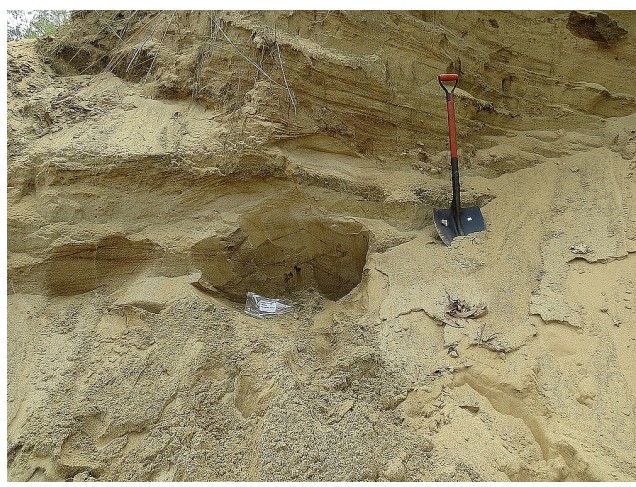

**Figure A12.** NGL - Cross-bedded medium-coarse sand unit ∼10 m above modern channel. Base of unit (S1) OSL dated at 14,000±3,000 years by Sinha et al. (2010).



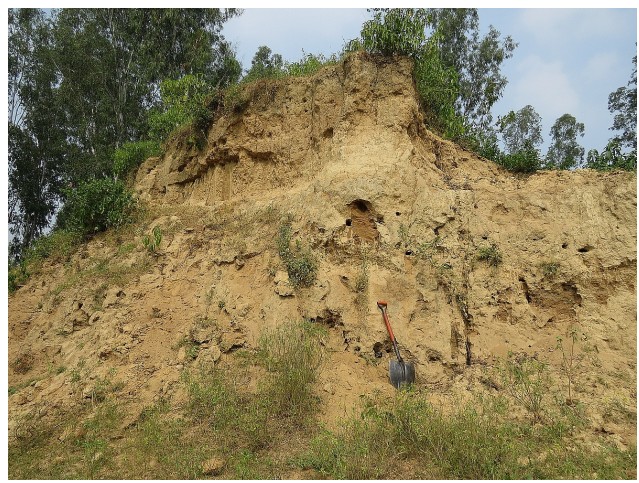

**Figure A13.** NGT - 4 m high exposure of low angle cross-bedded sands, topped with finer silt and mud deposits. Corresponds to OSL sample from this part of unit dated at 7,200±2,000 years by Sinha et al. (2010).





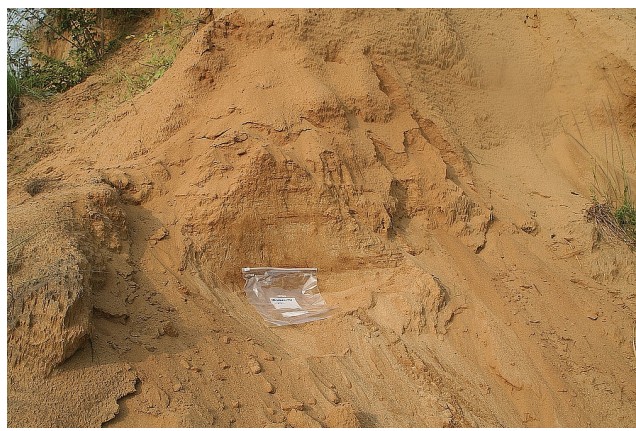

**Figure A14.** LH - Cross-bedded sand exposure (4 m high). Sample taken 2.2 m from top of exposure. Corresponds to OSL sample from unit dated at 23,500±1,500 years by Verma (2016).



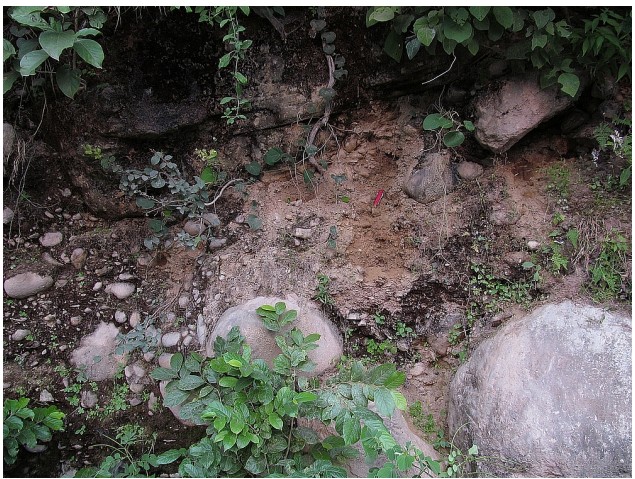

**Figure A15.** RLB - ∼42 m above modern channel on roadside cut. Poorly sorted, structureless fluvial conglomerate. Large, rounded boulders, cobbles and sands (Ray and Srivastava, 2010). 90 mm long penknife for scale.




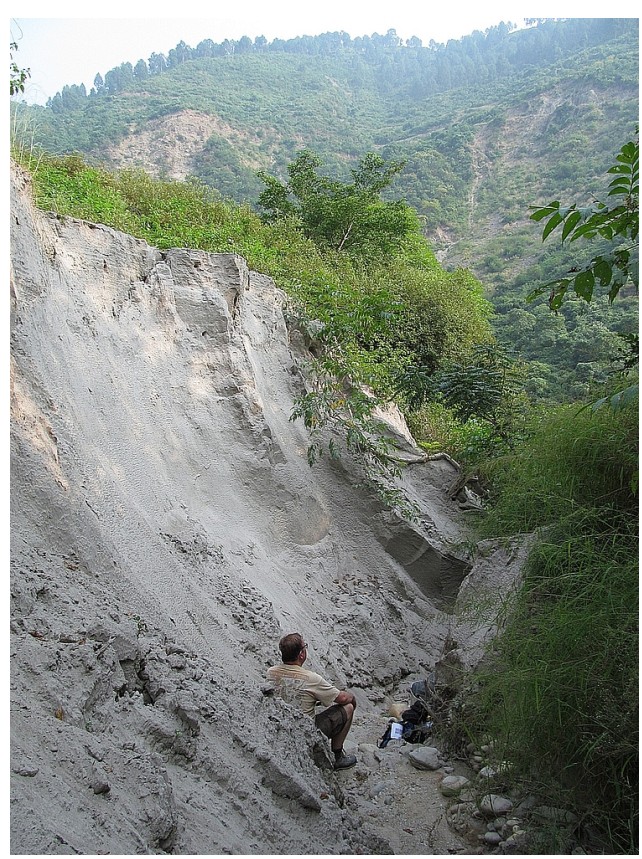

**Figure A16.** DV2013 - Laminated sand deposit ∼5 to 10 m thick formed in single event following the 2013 Alaknanda flooding.

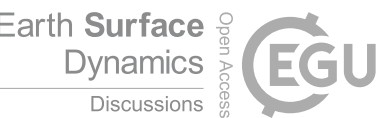

**Table 1.** CRN sample details, [10]Be concentrations and modelled erosion rates. Full sample details are given in Appendix A.

| Sample | Locality | Sampling date | Lat. | Lon. | Basin area (km²) | Mean basin elevation (m) | Sample age (years) | Age reference | Sample elevation (m, from DEM) | Average shielding factor* | Sample depth (cm) | Sample [10]Be concentration (×10³ at g⁻¹) | [10]Be concentration at time of deposition (×10³ at g⁻¹) | CAIRN-derived erosion rate (mm yr⁻¹) |
|---|---|---|---|---|---|---|---|---|---|---|---|---|---|---|
| BGM | Bagwan - modern | 06-Oct-2014 | 30.2255 | 78.6823 | 10,920 | 3,825 | Modern | n/a | 498 | 0.868 | 0 | 13.57±1.40 | 13.56±1.40 | 1.71±0.31 |
| BG1.8 | Bagwan - terrace | 06-Oct-2014 | 30.2253 | 78.6812 | 10,920 | 3,825 | 217±76 | Wasson et al. (2013) - OSL | 504 | 0.868 | 500 | 40.70±2.69 | 40.69±2.69 | 0.57±0.10 |
| DV2013 | Devprayag - 2013 flood | 05-Oct-2014 | 30.1499 | 78.6136 | 11,052 | 3,805 | 1 | n/a | 492 | 0.868 | 0 | 16.07±3.55 | 16.06±3.55 | 1.44±0.26 |
| DVTT2 | Devprayag - terrace | 05-Oct-2014 | 30.1508 | 78.6107 | 11,052 | 3,805 | 14,000±2,000 | Srivastava et al. (2008) - OSL | 530 | 0.868 | 600 | 7.09±2.45 | 6.66±2.45 | 3.48±1.02 |
| DVMT2 | Devprayag - terrace | 05-Oct-2014 | 30.1508 | 78.6153 | 11,052 | 3,805 | 10,000±2,000 | Ray and Srivastava (2010) - OSL | 517 | 0.868 | 650 | 14.69±1.22 | 14.27±1.22 | 1.63±0.29 |
| DVDF | Devprayag - terrace | 06-Oct-2014 | 30.1253 | 78.5905 | 18,716 | 3,870 | 10,000±2,000 | Ray and Srivastava (2010) - OSL | 559 | 0.868 | 1,300 | 23.19±1.28 | 23.04±1.28 | 1.01±0.18 |
| RLB | Rishikesh - terrace | 03-Oct-2014 | 30.1305 | 78.3322 | 21,675 | 3,670 | 6,940±650 | Sinha et al. (2010) - OSL | 393 | 0.868 | 300 | 15.61±1.27 | 14.52±1.27 | 1.60±0.29 |
| RFLO | Rishikesh - 2013 flood | 03-Oct-2014 | 30.1328 | 78.3342 | 21,675 | 3,670 | 1 | n/a | 370 | 0.868 | 20 | 12.86±1.58 | 12.85±1.58 | 1.81±0.33 |
| RAE1 | Raewalla - terrace | 08-Oct-2014 | 30.0053 | 78.2195 | 23,030 | 3,580 | 2,600±500 | Wasson et al. (2013) - OSL | 308 | 0.877 | 100 | 17.51±1.04 | 14.07±1.31 | 1.52±0.27 |
| RAE2 | Raewalla - terrace | 08-Oct-2014 | 30.0053 | 78.2195 | 23,030 | 3,580 | 1,000±200 | Wasson et al. (2013) - OSL | 308 | 0.877 | 80 | 20.76±1.09 | 19.08±1.28 | 1.12±0.20 |
| RAEM | Raewalla - modern | 08-Oct-2014 | 30.0054 | 78.2227 | 23,030 | 3,580 | Modern | n/a | 303 | 0.885 | 0 | 15.53±1.07 | 15.52±1.07 | 1.29±0.23 |
| CDT3 | Chandi Devi - terrace | 03-Oct-2014 | 29.9461 | 78.1757 | 23,221 | 3,560 | 9,760±1,040 | Sinha et al. (2010) - OSL | 309 | 0.877 | 320 | 14.19±1.11 | 12.91±1.12 | 1.66±0.30 |
| CDT4 | Chandi Devi - terrace | 03-Oct-2014 | 29.9398 | 78.1788 | 23,221 | 3,560 | 11,080±1,960 | Sinha et al. (2010) - OSL | 389 | 0.877 | 1,000 | 49.72±8.96 | 49.65±8.96 | 0.43±0.08 |
| GAPUB | Haridwar - modern | 11-Oct-2014 | 29.9067 | 78.1635 | 23,221 | 3,560 | Modern | n/a | 271 | 0.886 | 0 | 17.70±1.42 | 17.70±1.42 | 1.12±0.20 |
| LH | Landhaura - terrace | 07-Oct-2014 | 29.8105 | 77.9460 | 23,941 | 3,510 | 23,500±1,500 | Verma (2016) - OSL | 256 | 0.879 | 220 | 15.65±1.21 | 8.06±1.31 | 2.60±0.49 |
| NGL | Nagal - terrace | 07-Oct-2014 | 29.6698 | 78.1786 | 23,941 | 3,510 | 14,000±3,000 | Sinha et al. (2010) - OSL | 249 | 0.889 | 1260 | 19.07±1.13 | 18.86±1.13 | 1.03±0.19 |
| NGM | Nagal - terrace | 07-Oct-2014 | 29.6652 | 78.1850 | 23,941 | 3,510 | 7,200±2,000 | Sinha et al. (2010) - OSL | 258 | 0.889 | 850 | 16.67±1.28 | 16.49±1.28 | 1.18±0.21 |
| NGL | Nagal - terrace | 07-Oct-2014 | 29.6649 | 78.1859 | 23,941 | 3,510 | 7,200±2,000 | Sinha et al. (2010) - OSL | 259 | 0.889 | 250 | 18.96±1.36 | 17.27±1.44 | 1.12±0.20 |
| LUP09** | Rishikesh - modern | 11-Aug-2009 | 30.127 | 78.330 | 21,690 | 3,150 | Modern | n/a | 357 | 0.868 | 0 | 9.20±1.0 | n/a | 2.52±0.45 |

* Average shielding factor is the average of the combined shielding factors; topographic, snow and self-shielding values. These were calculated using a depth integrated approach (see Mudd et al., 2016).

** Details for this sample (BR924) are from Table 1 in Lupker et al. (2012). We have recalculated the erosion rate using the CAIRN method (Mudd et al., 2016).

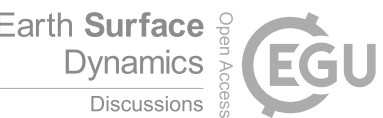

**Table 2.** Catchment area, average elevation and average [10]Be surface production rate for sub-catchments in the Ganga catchment

|  | Catchment area ($km^2$) | Catchment-average elevation (m) | Surface production rate (atoms $g^{-1}$ $yr^{-1}$) |
|---|---|---|---|
| Sub-catchment 1 | 1,955 | 1,606 | 11.08 |
| Sub-catchment 2 | 4,635 | 4,716 | 56.02 |
| Sub-catchment 3 | 1,801 | 5,033 | 70.51 |
| Sub-catchment 4 | 1,449 | 1,642 | 24.28 |
| Sub-catchment 5 | 169 | 4,483 | 49.13 |
| Sub-catchment 6 | 181 | 1,868 | 12.82 |
| Sub-catchment 7 | 253 | 1,404 | 9.57 |
| Sub-catchment 8* | 39 | 4,806 | 49.61 |
| Ganga (whole) | 23,038 | 3,560 | 33.16 |

*This sub-catchment represents the area upstream of Kedarnath during the 2013 Alaknanda flooding



**Table 3.** Default and range of parameter values used in numerical analysis

| Parameter | Default value | Range of modelled values |
|---|---|---|
| Landslide depth (m) | 2 | 0.5 - 5.0 |
| Catchment area (km$^2$) | 23,000 | - |
| % of catchment impacted by landsliding | 0.5 | - |
| Catchment-averaged surface production rate (atoms g$^{-1}$ yr$^{-1}$) | 35 | - |
| Background erosion rate (mm yr$^{-1}$) | 0.5 | 0.2-2.0 |
| Landslide surface production rate (atoms g$^{-1}$ yr$^{-1}$) | 35 | 10-60 |

*This sub-catchment represents the area upstream of Kedarnath during the 2013 Alaknanda flooding

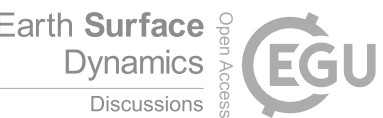

**Table 4.** Parameter values used to examine the difference in CRN-derived and volumetric sediment fluxes. More details are provided in Table S4 in Supplementary Material.

| Parameters | Scenario 1 | Scenario 2 |
|---|---|---|
| Catchment area ($km^2$) | 23,000 | 23,000 |
| % of catchment impacted by landsliding | 0.5 | 1.0 |
| Background erosion rate (mm $yr^{-1}$) | 0.7 | 0.8 |
| Average landslide depth (m) | 2.0 | 2.5 |
| Landslide surface production rate (atoms $g^{-1}$ $yr^{-1}$) | 35 | 45 |
| **Outlet concentration (atoms $g^{-1}$)** | **19,046** | **12,313** |
| **CRN-derived sediment flux (Mt)** | **68** | **105** |
| **Volumetric sediment flux (Mt)** | **87** | **204** |