# Peer review of "Temporal variability in detrital 10Be concentrations in a large Himalayan catchment"

_Earth Surface Dynamics, 2017_

## Referee Comment (RC1) · M. Lupker (Referee) · 17 Feb 2018

In this manuscript, Dingle et al. measured the cosmogenic 10Be concentration of sediments exported from the Ganga catchment in the west-Himalaya. Their data-set combines modern river sediments with floodplain and terraces deposits spanning the last ca. 25kyrs to investigate the 10Be concentration variability in such an actively eroding system. Variability in modern 10Be signals had already been documented in such systems, but the present results are surprising in that the authors show that this variability has remained largely unchanged over such long-time spans and despite known changes in climate. The authors provided a sensitivity analysis of the plausible causes for the observed variability mainly in relation to landsliding and sediment evacuation. Their conclusions reach beyond the presented case study since it is implied that in

actively eroding landscapes 10Be concentrations and erosion rates could be decoupled. Overall, I think this is an interesting and thought-provoking addition to the 10Be literature that tries to go beyond the simple interpretation of cosmogenic nuclides as "erosion rate meters" and seeks to take into account the actual erosion processes. I would therefore recommend this manuscript for publication in E-surf but I also believe that some aspects would benefit from a more in-depth discussion in the frame of a moderate revision.

General remarks:

- I am convinced by the demonstration that landsliding and landslide characteristics (location, depth etc.) can drive a significant variability in the 10Be signal if this material is mobilised rapidly within the fluvial network. Where I am a bit puzzled however, is how the models and sensitivity calculations in this manuscript compare with, for instance, the work from Niemi et al. (2005) or Yanites et al., (2009). These two papers have a thorough treatment of the landslide impact on 10Be signals (landslides are spatially and temporally resolved and include a proper landslide frequency-size distribution) but suggest that for higher order catchments the bias towards under-estimated CRN-derived erosion rates compared to volumetric rates is limited. In this work it seems that this bias is at least a factor 2 (Figure 8d), if not much more and these biases are also emphasized in section 5.3. Is this a result of how the different models are set-up or is this linked for instance to the hypsometry of the catchment (and large range of surface production rates)? I think, that the authors should be more upfront in comparing their approach with these published studies and better discuss were the apparent different conclusion may come from.

- Maybe somewhat related, I also think that the steady state assumption and what it implies for the conclusions of this study should be addressed in more details. The 0.5% of landslide surface, suggests that the landslide recurrence time at a given point of the landscape is roughly 200 years (assuming that the landslide are randomly distributed over the entire catchments). The 10Be concentration profiles are therefore likely quite

far from steady state (depending on background denudation rates and production rate) and the overall 10Be concentration of eroded material much lower than expected. This may therefore well be quite a strong assumption.

- As I mentioned earlier, I find the discussion on how landslides in different sub-catchments can drive a high variability in the 10Be signals convincing but in comparison the role of sediment storage and transfer in limiting this variability quite short. It seems however that this is crucial in interpreting the data since without these dampening effects the expected variability would be much higher. Is there a way to provide a more quantitative approach to this part of the system? Since you can model the expected variability that is induced by landslides, could you for instance estimate the size of the buffer needed to filter this variability to within roughly a factor 2? The fact that this variability is preserved over such a long time-scale would suggest that this buffer capacity is a characteristic of this catchment.

- One of the important messages of the manuscript is that CRN-derived sediment fluxes likely underestimate actual volumetric sediment fluxes (and maybe by a significant amount). Our data of Lupker et al., (2012) suggests that 10Be fluxes appear similar to slightly larger than gauged fluxes for large catchments in central Nepal. This work therefore suggests that the actual long-term fluxes implied by the CRN data might actually be much larger than currently measured (gauged) fluxes if this bias is taken into account. I would be curious to have the authors opinion on whether this could be a sign of a recent decrease in sediment fluxes or just induced by a large uncertainty on both methods?

Minor comments:

- This might be wrong on my side but I would not speak of error when referring to the natural variability in the 10Be concentrations of the river sediments (e.g. abstract l.13) or when referring to uncertainty in measured data (e.g. l.5, p.3).

- The SLHL 10Be production rates that were used in CAIRN for the calculation should

be mentioned somewhere. On the same topic, it be better to stay consistent throughout the manuscript with the use of CAIRN and not change for the CRONUS calculator for a series of sub-catchment (l.8, p.9). I know CAIRN does not explicitly need to the catchment averaged production rate estimates but it must also compute these values across the catchments. This may also explain why the entire Ganga catchment in table 2 has a production rate of 33 at/g/yr but the rest is modelled with a production rate of 35 at/g/yr in table 3 & 4.

- p.6, l.26 and Figure 4: I would keep the original sample name instead of LUPK09 to make it easier to trace across publications: BR924.

- What is the rationale for choosing the sub-catchments of Figure 7 and l.5, p.9?

- The chosen 0.5% of landslide area applies to co-seismic landsliding but is probably high for inter-seismic landsliding. Why has a such a high co-seismic landsliding value been chosen?

- L.13, p.9: see also Gorkha landslide statistics in Roback et al., 2017 (Geomorphology).

- L.2, p.8: there is a typo: draw a clearer picture

- L.1-2, p. 12: Godard et al., 2012 (JGR-ES) have contradictory results suggesting high glacial erosion in the Marsyangdi.

- L.14, p.13: I did not understand where the 7 and 10% came from.

All suggestions are meant in a constructive way and are open for discussion. I hope they will contribute to further improve the manuscript.

Maarten Lupker - ETH Zürich (17.02.2018)

---

## Short Comment (SC1) · 20 Feb 2018

P2-L35 : variability through time (more than a decade) of 10Be concentrations in large Himalayan rivers as also been documented in the lower reach of the Marsyandi river, central Nepal. See figure 3 from Godard et al. 2012 (JGR -ES : 10.1029/2011JF002230).

---

## Referee Comment (RC2) · Anonymous Referee #2 · 27 Feb 2018

Review Dingle et al.: Temporal variability in detrital 10Be in large Himalayan catchments

Dingle et al. present eighteen 10Be concentrations and denudation rates from modern river sediments, flood plain and terrace deposits along the middle part of the Ganga river. The observed variability in nuclide concentrations and denudation rates is discussed mainly in the light of stochastic sediment input into the river system. The presented model simulations demonstrate that cosmogenic nuclide concentration cannot always be used to determine sediment flux.

General comments:

The manuscript is developed around the cosmogenic nuclide concentrations which

show a not easy to explain scatter. One problem in comparing these nuclide concentrations is that the nuclide concentrations are derived from spatially and temporally different sediment samples. The spatial variation should be investigated independently from temporal variation with a larger data set pf modern samples from different locations. The temporal variations should then be investigated with a comparison to a modern sample from a comparable location. In addition to this, the situation is further complicated as samples with different deposition ages may be affected by different denudation processes (e.g., glacial- dominated versus landslide-dominated). Therefore, a comparison of all data from modern to deposited river samples is difficult and should be disentangled. Another problem could arise from the comparison of nuclide concentrations rather than denudation rates. Assuming sediment from two catchments with the same denudation rate but different production rates (e.g., two tributaries sampled above their confluence) will have different nuclide concentrations. Even so the catchments are subjected to the same erosion processes the nuclide concentrations are different. Therefore, it would make sense to compare denudation rates rather than nuclide concentrations. Furthermore, a way to go could be that the manuscript is developed around a discussion of the presented model in comparison to Niemi et al., 2005 and Yanites et al., 2009. This discussion could explain nicely why findings in this study are different from others. Furthermore, there should be an attempt to integrated the presented cosmogenic data into the model findings. Would it be possible to select the investigated catchments for model calculations where there are also cosmogenic nuclide data available supporting the model findings? Addressing the above major suggestions could make a stronger manuscript to be published in Earth Surface Dynamics. At the time being, the writing is not very concise and the data treatment not rigorous. The manuscript needs major revision with special attention to details (see suggestions below) before publication in Earth Surface Dynamics can be recommended.

Suggestions for changes:

Title:

Not convinced if this is the right title for this study. The study's strength seems to be more the model simulation than the cosmogenic concentrations. If this title stays then if should be: "Temporal variability of detrital 10Be concentrations in a large Himalayan catchment: the Alaknanda/Ganges river"?

Abstract:

P1, L5: "….Âăconcentrations at the catchment outlet are relatively stable in time". The concentrations are generally invariant over time. In addition, the use of catchment outlet here and in the entire manuscript is somehow misleading. The reader most likely attributes to catchment outlet the delta. This should be clarified. P1, L11-13: Is the doubling of sediment delivery to the Bay of Bengal just during 11 to 7 ka or does the doubling start at ∼9 ka and lasts until present-day? Again, the use of Ganga outlet is misleading.

1 Introduction

P2, L6-10: Simplification of this long sentence would be helpful for the understanding. Furthermore, please make sure of the consistent use of erosion or denudation rates. P2, L12: Would be helpful to clarify what the size of a small catchment is? P2, L25: Wondering why we jump to the Ganga-Brahmaputra delta. This delta is important, but not for the main findings of this manuscript. This paragraph needs to be packaged differently. P2, 28-29: Would this make sense?: "…..major Himalayan river systems has halved due to the reduction in monsoon rainfall since the early Holocene time." P3, L12L: Would it make sense to start this paragraph with a short introduction to the Alaknanda and Ganga rivers? P3, L14: What is meant by "ancient"? P3, L16: " making it the ideal techniques….". What is meant by it? Please be more concise.

2 Study area and context

Would It make sense to split this chapter up in 2.1 Study area and climate and 2.2 Sample information? P3, L26: Would it make sense to state here that the upper Ganga

catchment or the mountainous part of the catchment is investigated in this manuscript? P4, L2: The use of the abbreviation ISM is not explained. This should happen here. Please also cross-check the consistent use of other abbreviations.

3 Methods

A reorganization of this chapter could clarify the understanding of the method used. A possible way to go could be: 3.1 Sample collection (P5, L15-31) 3.2 Sample preparation (P5, 32 to P6, L14) 3.3 Calculation of denudation rates (P6, L15-24)

Alternatively, if the authors decide to structure the manuscript around the model simulation, the used model should also be described here.

This chapter needs to be treated with care for correct wording, for instance: P6. L13: Different value of half-life than used above. P6, L15-24: How was the glaciation taken into account? How are the shielding factors calculated?

4 Results

Results such as nuclide concentrations and denudation rates should clearly be separated from discussion (e.g. 2nd paragraph in the results should go into discussion). In addition, some results of the discussion addressing the model simulations could come in here if 3 Methods includes the model set up. P6, L26: What is with the third modern river sample? It should be mentioned here too. P7, L9: Is the sample BG1.8 not attributed a depth of 500 cm (see Table 1)?

5 Impact of stochastic inputs on CRN variability and sediment flux estimates

As mentioned above, if could make sense to describe the model simulation in 3 Methods. This would simplify the discussion. The discussion could be arranged in (as a suggestion): 5.1 Discussion of nuclide concentrations and denudation rates 5.2 Findings of model simulations (e.g., all simulations) 5.3 Sources of variability of CRN concentrations 5.4 Suitability of CRN as a proxy for sediment flux in large catchments

Figures:

Fig. 1: It would be helpful to Indicate in the figure what figure the red box refers to. Could the river names be added to this figure?

Fig. 3: Denudation rates to the sample ID would be helpful here. It is difficult to combine Table 1 with this map. Where is the LUP09 sample situated?

Fig. 4: Would it make sense to include also a figure with denudation rate versus age?

Fig. 9: Is the % grain size of the entire sample volume or from the sand/silt/clay volume? What are the references to source material? Y-axis label should be "% sand (gran size <1 mm)".

Fig. 10: Not totally clear what "CRN concentration (% of surface)" means. Could you clarify?

Tables:

Table 1: There is important information missing for the cosmogenic nuclide method. It could make sense to make two tables out of this table: Table 1 including the geomorphic and other information (e.g., column 1 to 9 plus grain size distribution by Dingle et al., 2016) and Table 2 column 10 to 15 plus additional information (e.g., analyzed grain size. production rates, apparent age). It is not clear what the average shielding factor includes. Why to the first eight samples have the same value in the shielding factor?

Table 4: Not sure if I missed something but Is scenario 1 and 2 the same as model A and B in figure 10? Please clarify.

[Figure]

---

## Author Comment (AC1) · 20 Apr 2018

Thanks very much for bringing this article to our attention. We have added a couple of statements regarding this study into our revised manuscript.

---

## Author Comment (AC2) · 20 Apr 2018

**Reviewer 1 (Maarten Lupker):**

In this manuscript, Dingle et al. measured the cosmogenic 10Be concentration of sediments exported from the Ganga catchment in the west-Himalaya. Their data-set combines modern river sediments with floodplain and terraces deposits spanning the last ca. 25kyrs to investigate the 10Be concentration variability in such an actively eroding system. Variability in modern 10Be signals had already been documented in such systems, but the present results are surprising in that the authors show that this variability has remained largely unchanged over such long-time spans and despite known changes in climate. The authors provided a sensitivity analysis of the plausible causes for the observed variability mainly in relation to landsliding and sediment evacuation. Their conclusions reach beyond the presented case study since it is implied that in actively eroding landscapes 10Be concentrations and erosion rates could be decoupled. Overall, I think this is an interesting and thought-provoking addition to the 10Be literature that tries to go beyond the simple interpretation of cosmogenic nuclides as "erosion rate meters" and seeks to take into account the actual erosion processes. I would therefore recommend this manuscript for publication in E-surf but I also believe that some aspects would benefit from a more in-depth discussion in the frame of a moderate revision.

General remarks: - I am convinced by the demonstration that landsliding and landslide characteristics (location, depth etc.) can drive a significant variability in the 10Be signal if this material is mobilised rapidly within the fluvial network. Where I am a bit puzzled however, is how the models and sensitivity calculations in this manuscript compare with, for instance, the work from Niemi et al. (2005) or Yanites et al., (2009). These two papers have a thorough treatment of the landslide impact on 10Be signals (landslides are spatially and temporally resolved and include a proper landslide frequency-size distribution) but suggest that for higher order catchments the bias towards under-estimated CRN-derived erosion rates compared to volumetric rates is limited. In this work it seems that this bias is at least a factor 2 (Figure 8d), if not much more and these biases are also emphasized in section 5.3. Is this a result of how the different models are set-up or is this linked for instance to the hypsometry of the catchment (and large range of surface production rates)? I think, that the authors should be more upfront in comparing their approach with these published studies and better discuss were the apparent different conclusion may come from.

Firstly, thank you for your comments on our manuscript. In response to your first general remark, we believe that the main differences lie in the fact that the analysis we have carried out examines the effect of the largest events in a catchment, rather than simulate how catchment-averaged concentrations vary in a landscape that has been allowed to evolve towards steady state conditions. Furthermore, we are not trying to compare or fit our analysis to actual data such as we see in the Niemi *et al.* (2005) paper. Instead, we want to test how outlet concentrations vary in response to a large single event with variable characteristics. This allows us to explore if there are there certain conditions under which deeper landslides with lower CRN concentration sediment might have a greater impact. We also did not add a power-law distribution of landslide area/depths or have a landscape evolution style model, which would allow stochastic landslides to continually occur over the landscape until nuclide concentrations attain some kind of 'steady-state'. Instead we apply a background erosion rate (0.2-2 mm/yr) across the landscape (which we assume to represent this steady state condition) and we add in an extreme event (which generates numerous landslides) to see what impact this has. We use an average landslide depth for this extreme event.

One could argue that this may be unrepresentative, given the power law distribution of landslide size/depth vs frequency. But, at any one time it is unlikely that a single concentration sample would integrate sediment from this full distribution of events. Instead, it seems more realistic that a catchment-averaged sample is likely to integrate a proportion of that distribution. As such, we vary our average landslide depth to represent the variation in the position of that window along the power law distribution. We assume that the smaller and more frequent events (we could consider these the inter-seismic, monsoon storm initiated landslides) are captured in the background erosion rate. The event we model is the equivalent of adding in events from the high magnitude tail end of a power law distribution that might record a seismic trigger.

Without jumping ahead too much to our responses to other comments, we have perhaps under-represented a couple of important points; namely, the assumption that the landslide material is generated from a landscape under steady-state conditions (i.e. with a fully developed concentration profile), and that following a landslide, there is instantaneous delivery of this sediment to the fluvial network. By incorporating these two factors into our modelling, we can produce additional results that are much more comparable to the Niemi *et al.* (2005) and Yanites *et al.* (2009) papers which we feel strengthens this manuscript.

Abstract: "We demonstrate that in certain systems…..it is possible to generate larger sediment fluxes…."

- Maybe somewhat related, I also think that the steady state assumption and what it implies for the conclusions of this study should be addressed in more details. The 0.5% of landslide surface, suggests that the landslide recurrence time at a given point of the landscape is roughly 200 years (assuming that the landslide are randomly distributed over the entire catchments). The 10Be concentration profiles are therefore likely quite far from steady state (depending on background denudation rates and production rate) and the overall 10Be concentration of eroded material much lower than expected. This may therefore well be quite a strong assumption.

Yes, we agree that this is a strong assumption. This is why we have examined how varying the initial surface concentration of landsliding changes the outlet concentration (see Figure 8 part c). On page 10 we state the assumptions we have made about this as well. We also feel that this better represents the fact that landslides are more likely to occur in parts of the landscape with faster background erosion rates (e.g. Binnie *et al*., 2007), and therefore, lower surface concentrations.

However, we agree in that we have perhaps over-looked this, and have run additional analyses to examine how this might influence our results. An additional set of runs have been made using lower landslide surface CRN production rates of 10 atoms/g/yr (rather than 35), such that sediment generated by the landslides has a lower (depth-averaged) CRN concentration than initial model runs. The results of this can be seen below.

[Figure]

The absolute CRN concentrations we get here are much lower than the initial runs (maximum concentration in Fig 8 is ~72,000 atoms/g where average landslide depths are 0.5m), which also helps bring the volumetric and CRN-derived sediment fluxes to more comparable levels in the figure on the right (more so where background erosion rates are higher). This also suggests that the effect of rare but large magnitude events in systems which are more stable with lower background erosion rates will have a greater effect on catchment-averaged CRN concentrations.

- As I mentioned earlier, I find the discussion on how landslides in different subcatchments can drive a high variability in the 10Be signals convincing but in comparison the role of sediment storage and transfer in limiting this variability quite short. It seems however that this is crucial in interpreting the data since without these dampening effects the expected variability would be much higher. Is there a way to provide a more quantitative approach to this part of the system? Since you can model the expected variability that is induced by landslides, could you for instance estimate the size of the buffer needed to filter this variability to within roughly a factor 2? The fact that this variability is preserved over such a long time-scale would suggest that this buffer capacity is a characteristic of this catchment.

Yes, we think this is also something that is worthwhile examining. We have included an additional plot showing what happens to the catchment outlet concentration when different proportions of the landslide flux are delivered. We have explored what happens if 20,10,5 and 3% of the landslide generated sediment is mixed into the catchment average, respectively. We examine only the first year, as would expect the input of landslide material to decrease in subsequent years.

[Figure]

In the figure on the left, we have reduced the landslide event surface production rate to 10 atoms/g/yr to mimic regions with faster background erosion rate, and then used two catchment-wide background erosion rates (for 'non-event' parts of the catchment) (0.6 and 2.0 mm/yr). What we see is that under faster background erosion rates, the magnitude of landsliding event can be 'lost' in expected variability (i.e. all values are within 100% of the highest concentration) if only 5% of the landslide material makes it into the fluvial network. Under lower background erosion rates, to reduce all concentrations within ~100% of the largest concentration, a maximum of ~3% of the landslide material needs to be entrained). If greater quantities of landslide material get into the network, the catchment-averaged concentrations become much lower (i.e. beyond what you might expect within natural variability) with deeper landsliding events. By incorporating this into our calculations, we see that the relationship between volumetric and CRN-derived sediment fluxes is much more comparable (figure on the right).

This raises some interesting points. The valley fill estimate for the Ganga basin in Blothe and Korup (2012) was pretty low in comparison to other Himalayan systems, and we see a fairly limited sand content in modern gravel bars (~10%). It is possible that a lot of this sediment is being stored within the landslide deposits themselves, and is only being mobilised and transported through the system by very high (and probably very localised) discharge events (e.g. a monsoonal storm or GLOF). It looks as if the sediment generated by these types of events is capable of driving significant change, but its impact is limited by the ability of the fluvial system to mobilise it. One explanation is that during strong monsoon seasons or discharge events, a greater proportion of low CRN concentration sediment

is mobilised from deposits/hillslopes as water stage rises and this could drive the variability we see at the mountain front.

- One of the important messages of the manuscript is that CRN-derived sediment fluxes likely underestimate actual volumetric sediment fluxes (and maybe by a significant amount). Our data of Lupker et al., (2012) suggests that 10Be fluxes appear similar to slightly larger than gauged fluxes for large catchments in central Nepal. This work therefore suggests that the actual long-term fluxes implied by the CRN data might actually be much larger than currently measured (gauged) fluxes if this bias is taken into account. I would be curious to have the authors opinion on whether this could be a sign of a recent decrease in sediment fluxes or just induced by a large uncertainty on both methods?

Our additional analysis suggests that this pattern may not be as apparent as we first proposed. Our main concern with the gauged fluxes is that they are unlikely to fully capture/truly represent what is being transported during the big monsoonal storms, and they certainly don't record what is being moved as bedload. These 2-3 day events seem to be really key in moving sediment out of these systems and into the Plain. These suspended sediment records are also pretty intermittent and patchy in space too, and can be destroyed during large flows, so it's difficult to quantify the uncertainty. Similarly, the CRN records might not be incorporating shorter-term trends or changes.

Minor comments: - This might be wrong on my side but I would not speak of error when referring to the natural variability in the 10Be concentrations of the river sediments (e.g. abstract l.13) or when referring to uncertainty in measured data (e.g. l.5, p.3).

Yes agreed. This has been changed to 'variability' and 'uncertainty', respectively.

- The SLHL 10Be production rates that were used in CAIRN for the calculation should be mentioned somewhere. On the same topic, it be better to stay consistent throughout the manuscript with the use of CAIRN and not change for the CRONUS calculator for a series of sub-catchment (l.8, p.9). I know CAIRN does not explicitly need to the catchment averaged production rate estimates but it must also compute these values across the catchments. This may also explain why the entire Ganga catchment in table 2 has a production rate of 33 at/g/yr but the rest is modelled with a production rate of 35 at/g/yr in table 3 & 4.

CAIRN doesn't explicitly produce an average production rate, you would need to average all of the values in the production rasters for each sample. The 35 at/g/yr in table 2 is just an average value we have used in the calculations.

- p.6, l.26 and Figure 4: I would keep the original sample name instead of LUPK09 to make it easier to trace across publications: BR924.

Yes, changed.

- What is the rationale for choosing the sub-catchments of Figure 7 and l.5, p.9?

This was to generate a range of different catchments with the maximum range of possible production rates, yet maintain comparable drainage areas between some of the examples. The main purpose was to essentially demonstrate that events could happen in different sub-catchments with very different CRN production rates.

- The chosen 0.5% of landslide area applies to co-seismic landsliding but is probably high for inter-seismic landsliding. Why has a such a high co-seismic landsliding value been chosen?

This is simply based on the landslide areas generated by the Chi-Chi and Gorkha earthquakes (~87-128 km$^2$), and expressed as a fraction of the total Ganga catchment (which equates to ~115 km$^2$). This is just to reflect the area of landsliding which can realistically be generated by an extreme event,

although we are aware that this may be slightly on the high side, as would be unlikely that all of the landsliding would occur within a single catchment.

- L.13, p.9: see also Gorkha landslide statistics in Roback et al., 2017 (Geomorphology).

Reference added

 - L.2, p.8: there is a typo: draw a clearer picture –

Thanks – changed!

L.1-2, p. 12: Godard et al., 2012 (JGR-ES) have contradictory results suggesting high glacial erosion in the Marsyangdi.

Yes. This is an important reference we hadn't seen.

L32 p3. We have added in a reference to Godard *et al.* (2012) concerning geomorphic process domains

L9-13 p13 . Changed the reference to Heimsath and McGlynn to look at 'localised' erosion and added the following text:

'An analysis of the evolution of detrital $^{10}$Be concentrations along the Marsyandi River suggested that low concentration $^{10}$Be inputs from glaciated tributaries dilute main stem $^{10}$Be concentrations (Godard *et al.,* 2012). In this instance, glacial erosion was averaged at ~5 mm/yr in the High and Tethyan Himalayan portions of the Marsyandi catchment, suggesting that glacially derived sediments may complicate detrital CRN concentrations and interpretation of catchment-averaged denudation rates.'

- L.14, p.13: I did not understand where the 7 and 10% came from.

We did not explain this clearly. This is to mimic the effect of buffering of event concentrations, which we know must happen across these systems. We have also removed this section from the manuscript now as buffering is covered in more detail in the modelling work which is sufficient to help explain the Holocene climate change.

All suggestions are meant in a constructive way and are open for discussion. I hope they will contribute to further improve the manuscript. Maarten Lupker - ETH Zürich (17.02.2018)

**Reviewer 2**

Dingle et al. present eighteen 10Be concentrations and denudation rates from modern river sediments, flood plain and terrace deposits along the middle part of the Ganga river. The observed variability in nuclide concentrations and denudation rates is discussed mainly in the light of stochastic sediment input into the river system. The presented model simulations demonstrate that cosmogenic nuclide concentration cannot always be used to determine sediment flux.

General comments: The manuscript is developed around the cosmogenic nuclide concentrations which show a not easy to explain scatter. One problem in comparing these nuclide concentrations is that the nuclide concentrations are derived from spatially and temporally different sediment samples.

Yes we agree that this is a challenge with these types of data and this is why we wanted to explore this subject. In many cases, this is the only type of data we have available as there are very few long-term sampling campaigns.

The spatial variation should be investigated independently from temporal variation with a larger data set pf modern samples from different locations. The temporal variations should then be investigated with a comparison to a modern sample from a comparable location.

This is a worthy aspiration, and makes good sense. However, building a temporal record is limited by having well preserved, dateable and sufficiently thick terrace deposits to ensure that the sample has been sufficiently shielded since deposition. In fact, some of our initial sampling was aimed at this, but variable quartz concentrations made analyses difficult. The data we have presented do offer a valuable insight based on terraces preserved close to the mountain front along the Ganga River. We agree that this could be the basis of a future and longer term monitoring campaign. The majority of samples are also below the final major tributary confluence in the Ganga catchment (Alaknanda-Baghirati) so the spatial area these samples represent should be fairly comparable when considered against the total size of the Ganga catchment.

In addition to this, the situation is further complicated as samples with different deposition ages may be affected by different denudation processes (e.g., glacial- dominated versus landslide-dominated). Therefore, a comparison of all data from modern to deposited river samples is difficult and should be disentangled.

We agree that different denudation processes complicate this signal, which is what we try to explore in the paper. What we are essentially trying to ask is 'what does an average CRN concentration actually represent in terms of sediment generation and transport?' By plotting against age, we hope that it is possible to appreciate the likely changing influence of glacial debris.

Another problem could arise from the comparison of nuclide concentrations rather than denudation rates. Assuming sediment from two catchments with the same denudation rate but different production rates (e.g., two tributaries sampled above their confluence) will have different nuclide concentrations. Even so the catchments are subjected to the same erosion processes the nuclide concentrations are different. Therefore, it would make sense to compare denudation rates rather than nuclide concentrations.

See above comment. The point is we measure CRN concentrations from samples, not denudation rates. We calculate denudation rates based on a series of assumptions applied to these concentrations. This is why we are sticking with concentrations, although have also considered denudation rates calculated using the CAIRN method (Mudd *et al*., 2016) in Figure 6.

Furthermore, a way to go could be that the manuscript is developed around a discussion of the presented model in comparison to Niemi et al., 2005 and Yanites et al., 2009. This discussion could explain nicely why findings in this study are different from others. Furthermore, there should be an attempt to integrated the presented cosmogenic data into the model findings.

Agreed. Please see response to Reviewer 1.

Would it be possible to select the investigated catchments for model calculations where there are also cosmogenic nuclide data available supporting the model findings?

Good idea, but no, our data are limited to a small spatial area close to the mountain front and therefore integrate the majority of the Ganga catchment. Data further upstream are limited.

Suggestions for changes:

Title: Not convinced if this is the right title for this study. The study's strength seems to be more the model simulation than the cosmogenic concentrations. If this title stays then if should be: "Temporal variability of detrital 10Be concentrations in a large Himalayan catchment: the Alaknanda/Ganges river"?

We feel the strength of this paper is the new dataset we have presented, and the modelling work is more of an investigation/sensitivity analysis to explore possible explanations. Will modify to "Temporal variability in detrital 10Be concentrations in a large Himalayan catchment"

Abstract:
P1, L5: ". . ..Âaconcentrations at the catchment outlet are relatively stable in time". The ˘ concentrations are generally invariant over time.

Ok, changed.

In addition, the use of catchment outlet here and in the entire manuscript is somehow misleading. The reader most likely attributes to catchment outlet the delta. This should be clarified.

We have clarified what we term as outlet at the end of the introduction and removed references specifically to the Ganga outlet prior to this.

"Motivated by the results, we examine the impact of stochastic inputs of sediment from the upstream mountain catchment on 10Be concentrations close to the mountain front (herein termed the Ganga outlet)."

P1, L11-13: Is the doubling of sediment delivery to the Bay of Bengal just during 11 to 7 ka or does the doubling start at ~9 ka and lasts until present-day? Again, the use of Ganga outlet is misleading.

See above. We have also clarified this sentence to read 'doubling of sediment delivery to the Bay of Bengal between 11-7 ka'.

1 Introduction
P2, L6-10: Simplification of this long sentence would be helpful for the understanding. Furthermore, please make sure of the consistent use of erosion or denudation rates.

This sentence has been split into two.

"Based on this approach, catchment-averaged denudation rates can be calculated, and converted into CRN-derived sediment fluxes which are typically averaged over hundred to thousand year timescales. These timescales are a function of the landscape denudation rate (i.e. the time taken to erode to a depth equivalent to the cosmic ray attenuation length in that landscape)."

Erosion changed to denudation

P2, L12: Would be helpful to clarify what the size of a small catchment is?

<100 km$^2$ (i.e. Yanites *et al*., 2009). This has been added to the sentence.

P2, L25: Wondering why we jump to the Ganga-Brahmaputra delta. This delta is important, but not for the main findings of this manuscript. This paragraph needs to be packaged differently.

The Ganga-Brahmaputra is the only long-term record we have of sediment flux out of the Himalayan mountains at $10^4$-$10^5$ yr timescales. This is vital in comparing sediment flux estimates (and therefore erosion rate estimates) where we know that the majority of sediment generated within the mountains bypasses the foreland basin and is delivered here. Understanding that there has been a big change in sediment delivery to the delta during the early Holocene underpins much of this paper, as we would expect to see this preserved in the 'erosion' record. Our results suggest that the variability in CRN concentrations is sufficiently high to mask these kinds of large-scale climatic shifts which we see preserved in the off-shore record.

P2, 28-29: Would this make sense?: ". . ...major Himalayan river systems has halved due to the reduction in monsoon rainfall since the early Holocene time."

Changed to "Sediment volumes in the Ganga-Brahmaputra delta imply that overall sediment flux from these two major Himalayan river systems has halved due to the reduction in monsoon rainfall since the early Holocene"

P3, L12L: Would it make sense to start this paragraph with a short introduction to the Alaknanda and Ganga rivers?

We thought it would be less confusing to keep all of this information for the following section (which is the next paragraph as well).

P3, L14: What is meant by "ancient"?

Samples which are not taken from the modern river channel (i.e. are preserved in terraces or flood deposits which we have independent OSL dates for).
Text changed to "…in both ancient (i.e. independently dated terrace and floodplain deposits) and modern fluvial sediments…"

P3, L16: " making it the ideal techniques. . ..". What is meant by it? Please be more concise.

Other isotopes with shorter half-lives may begin to decay over the timescales we are interested in, which is another factor we would then have to correct for.

2 Study area and context
Would It make sense to split this chapter up in 2.1 Study area and climate and 2.2 Sample information?

We have added in 'sample information' as a sub-heading.

P3, L26: Would it make sense to state here that the upper Ganga catchment or the mountainous part of the catchment is investigated in this manuscript?

We have added this text to clarify: "This study focuses on the portion of the Ganga catchment upstream of the Himalayan mountain front, the most downstream extent of which we also term the catchment outlet."

P4, L2: The use of the abbreviation ISM is not explained. This should happen here. Please also cross-check the consistent use of other abbreviations.

Done. P3 L21 is first use of the term so it is explained here.

3 Methods
A reorganization of this chapter could clarify the understanding of the method used. A possible way to go could be: 3.1 Sample collection (P5, L15-31) 3.2 Sample preparation (P5, 32 to P6, L14) 3.3 Calculation of denudation rates (P6, L15-24) Alternatively, if the authors decide to structure the manuscript around the model simulation, the used model should also be described here.

We have added in sub-section headings of Sample collection, Sample preparation and Denudation rate calculations.

This chapter needs to be treated with care for correct wording, for instance: P6. L13: Different value of half-life than used above.

Corrected the first instance to 1.36 x 10$^6$ years

P6, L15-24: How was the glaciation taken into account? How are the shielding factors calculated?

Glacial cover was determined from the GLIMS database and fed into the CAIRN modelling as a raster. Shielding factors are calculated in the CAIRN model/methodology (Mudd *et al*., 2016) – this is not something to be explained in this manuscript as details are available in published material which is referenced throughout this paragraph.

4 Results
Results such as nuclide concentrations and denudation rates should clearly be separated from discussion (e.g. 2nd paragraph in the results should go into discussion). In addition, some results of the discussion addressing the model simulations could come in here if 3 Methods includes the model set up.

Much of the second paragraph from Results has now been moved into the discussion in a new subsection called 'CRN sample interpretation'.

P6, L26: What is with the third modern river sample? It should be mentioned here too.

The sample we have referred to as LUPK09 (which has now been renamed BR924) refers to a sample presented in Lupker *et al*. (2012) taken at the same location as our sample RAEM. We have one additional sample further upstream (BGM) which is further upstream of the Alaknanda-Bhagirathi confluence. As such it has been separated from the other samples in this section as does not integrate the Bhagirathi drainage area. A sentence has been added to this effect in the first paragraph of the results section.

P7, L9: Is the sample BG1.8 not attributed a depth of 500 cm (see Table 1)?

The sample was taken from 500cm below the modern surface, but the individual event bed was measured at ~50cm thick.

5 Impact of stochastic inputs on CRN variability and sediment flux estimates
As mentioned above, if could make sense to describe the model simulation in 3 Methods. This would simplify the discussion. The discussion could be arranged in (as a suggestion): 5.1 Discussion of nuclide concentrations and denudation rates 5.2 Findings of model simulations (e.g., all simulations) 5.3 Sources of variability of CRN concentrations 5.4 Suitability of CRN as a proxy for sediment flux in large catchments

We had actually considered this in an earlier draft of the manuscript but felt it was clearer to separate the numerical analysis from the CRN sampling. This was because the variability we observe in our samples prompted the numerical analysis. The degree of variability in the CRN samples and their insensitivity to Holocene climate was a finding in itself, so it was difficult to explain the numerical analysis without first presenting the CRN sample results (if that makes sense!).

Figures:
Fig. 1: It would be helpful to Indicate in the figure what figure the red box refers to. Could the river names be added to this figure?

Yes, added.

Fig. 3: Denudation rates to the sample ID would be helpful here. It is difficult to combine Table 1 with this map. Where is the LUP09 sample situated?

Position of LUP09 (now BR924) has been added. Denudation rates are shown in Figure 6 as well – have tried adding them onto this figure but over-clutters it.

Fig. 4: Would it make sense to include also a figure with denudation rate versus age?

Given that most of the samples fall within the variability of the modern samples (Fig. 6) it seems doubtful it would present anything extra to the argument.

Fig. 9: Is the % grain size of the entire sample volume or from the sand/silt/clay volume? What are the references to source material? Y-axis label should be "% sand (gran size < 1mm)".

The graph shows the fraction of the entire volumetric sample which is finer than 1 mm. Reference has been added to figure caption, and axis label updated to % sand (grain size < 1mm).

Fig. 10: Not totally clear what "CRN concentration (% of surface)" means. Could you clarify?

Yes it's simply the concentration at that depth represented as a fraction of the concentration at the surface (which will be the maximum value – i.e. 100%).

Tables:
Table 1: There is important information missing for the cosmogenic nuclide method. It could make sense to make two tables out of this table: Table 1 including the geomorphic and other information (e.g., column 1 to 9 plus grain size distribution by Dingle et al., 2016) and Table 2 column 10 to 15 plus additional information (e.g., analyzed grain size. production rates, apparent age). It is not clear what the average shielding factor includes. Why to the first eight samples have the same value in the shielding factor?

We stated the grain size analysed in the methods section (250-500 μm). The average shielding factor is an average of the topographic and snow shielding factors generated in CAIRN, which is stated as a footnote on the table. The repeated shielding factors have subsequently been corrected, and revised erosion rates added to Table 1 and Figure 4 and 5 (although these changes are very small).

Table 4: Not sure if I missed something but is scenario 1 and 2 the same as model A and B in figure 10? Please clarify.

No these are not the same values. Scenario 1 and 2 are designed to simply highlight that a doubling of volumetric sediment flux can be generated within background CRN variability, perhaps explaining why no obvious change in CRN concentrations are documented at the Ganga mountain front through the Holocene. We have subsequently removed the scenario 1 and 2 section as feel that the additional analysis looking at buffering explains the Holocene climate story.

---

## Referee Report (RR1)

This paper considers variability in *in situ* [10]Be concentration in one catchment draining most of the central Himalayas in order to understand how temporal variability in [10]Be concentration may affect estimated erosion rates. Overall I think the paper is an interesting contribution to add to empirical work previously done by Lupker and Gonzalez and modeling done by Niemi and Yanites. I have a few concerns about the paper that I think the authors could address fairly easily to make the paper acceptable for publication.

1) The authors have completely ignored the Earth Surface Processes and Landforms paper by Gonzalez that came out last year. In that paper Gonzalez and coauthors (full disclosure that I am the corresponding author on that paper) compare [10]Be (both *in situ* and meteoric) measurements of pairs of samples from western China and also do a meta-analysis of previously published replicate studies. Not including this paper is a major oversight because it means that it feels like Dingle is reinventing the wheel rather than engaging in conversation with other manuscripts considering similar topics. In particular, I think that considering similarities and differences to the Gonzalez et al dataset would be appropriate for the discussion and the legwork Gonzalez did in a global meta-analysis should contribute to the introduction.

2) I am bothered by the way that the manuscript is structured with respect to the landslide modeling. The manuscript has a standard set up of introduction, methods, results, discussion, conclusion for the [10]Be concentrations and derived denudation rates. However, the authors have thrown into the discussion section all of the background (introduction), methods, results, and discussion for their landslide model. I think that it would be more appropriate to introduce the model early, develop it in the methods, present the results in the results section, and discuss it in the discussion section.

3) Although the authors do an excellent job of addressing a number of possible reasons for the variability in isotopic data, they do not (as far as I can tell) mention the one process that Lupker concluded dominated in the Himalayas – varying sourcing from catchments with similar average erosion rates but different production rates due to varying elevation.

4) I'm confused about why you use concentration differences when you have location differences for your samples. It seems like in that case it is better to compare denudation rates (as Schaller did) rather than comparing concentrations, which are biased by variable upstream elevation. I agree that if you are comparing two samples at the exact sample (or nearly exact same) sample site, then comparing concentration is more appropriate, but given the variability in location, I think that comparing denudation may be better.

5) I would like to see some conclusions that you can draw that may help others to sample better. It's all well and good to say "be careful, the system doesn't work like you assume" but unless you can offer concrete suggestions, people will continue to happily run around and grab bags of sand and assume that everything is well averaged. What can you recommend we do?

In addition to these major points, I have a few minor comments:

Term use throughout: Why do you use CRN instead of [10]Be? You only measured *in situ* [10]Be, so it seems silly to refer to the generic term CRN throughout.

Acronyms throughout: Be sure to define all acronyms (CRN, CAIRN, CRONUS) at first appearance.

P5 L26: I would put the grain size you field sieved to here.

P6 L7: Can you justify that sometimes including this finer grain size doesn't affect your concentration measurements?

Section 3.2: I think it would be extremely helpful to have a table that has the quartz amount, carrier mass, and AMS measurements (ratios, blanks, blank-corrected ratios) so that calculations can be checked and replicated. These are standard supporting tables these days. This is especially important since your blank values range from 4 to 54% of your measured ratios.

Section 3.3: Be sure to include all the parameters that one needs to replicate your work (like assumed sediment density). It is also useful to include a table that one can easily input into an erosion rate calculator should they want to do a meta-analysis or recalculate your denudation rates as parameters change.

Figures: All figures have text that is hard to read. I assume it is a reproduction for review issue, but wanted to ensure that you check that text is easily legible in final publication format.

Figure 2: If you aren't using lat/long, you need to say what datum and projection your coordinates are from. Given that Table 1 has lat/long in decimal degrees, you may consider changing figure 2 to use decimal degrees as well.

Table 2: Why mean elevation instead of effective elevation? (And, I am assuming "average" is "mean" and not "median" or "mode". Probably would be good to clarify.)

This review is intended to be constructive and I would be happy to clarify or answer questions for the authors.

Amanda Schmidt

---

## Author Response (AR2)

SCHOOL of GEOSCIENCES
The University of Edinburgh
Drummond Street
Edinburgh EH8 9XP

Telephone +44 (0) 131 650 9170
Fax +44 (0)131 650 2524
elizabeth.dingle@ed.ac.uk

`

Michele Koppes
Associate Editor, Earth Surface Dynamics

27th June 2018

Dear Dr. Koppes,

     Thank you for considering our manuscript 'Temporal variability in detrital $^{10}$Be concentrations in a large Himalayan catchment'. We are grateful to both reviewers for again providing constructive feedback which we have incorporated into our revised manuscript and in particular we hope adds an interesting statistical element to our story.

As requested, we have mentioned how our work relates to that of Gomez *et al.* (2017), and have modified how we have represented our non-steady state landscapes in the modelling element of the manuscript. This does not notably change our results but, as noted by reviewer 1, is a more logical approach. We have also added new statistical analysis to better constrain the number of $^{10}$Be samples required to fully capture the variability observed in the Ganga River.

Please find below detailed responses to the individual points raised by both reviewers, along with a version of our manuscript highlighting changes we have made to each reviewer comment. We have endeavoured to address all concerns raised by the reviewers.

Yours sincerely,

Elizabeth Dingle
Corresponding author

**Reviewer 1 (Maarten Lupker):**

I just reviewed a revised manuscript by Elizabeth Dingle and co-workers on the variability of 10Be signals in fluvial sediments of the Ganga catchment in the central-western part of the Himalaya. As mentioned in my previous review I think this is a valuable contribution that should be considered for publication. Most of the comments that I made during my previous review have been addressed and the manuscript improved. The revision has induced some changes in the conclusion, notably on the magnitude of the difference between cosmogenic nuclide-derived denudation rates and actual erosion fluxes under landsliding conditions. I find these new conclusions more convincing, especially in comparison with other published literature on the subject.

I still have one main comment on the new approach that needs to be considered before publication. In

order to take into account that the cosmogenic nuclide profile being mobilised by landslide is likely not at steady state, the authors have chosen to model this by reducing the surface production rate by 10 at/g/yr (p11, l17 to 27 & Figure 9). First, it is unclear to me why this value has been chosen, second it seems to me that you would be better off considering a reduction of exposure duration but keep the initial production rate. Doing so, would be analogous to change the recurrence interval of landsliding at a given point of the landscape, a quantity that has a geomorphological meaning rather than considering the landslides to artificially occur at a lower elevation (what is done by lowering the production rate). This is unlikely to change the main conclusions of the paper but would, in my opinion, be a more logical approach in this case.

Yes, we can see your point here. Unfortunately, we do not have a temporal component to the model so reducing the exposure duration isn't possible. Instead, we have re-run using a higher erosion rate across the region of landsliding ('average landslide erosion rate'), as opposed to a uniform rate across the landscape with variable 10Be production rates. Hopefully this makes more sense – in regions which undergo more frequent landsliding, the erosion rate averaged across this region is also likely to be higher. Choosing the exact numbers is again difficult. We have applied an average landslide erosion rate (to the region of landsliding) of 3.0 mm/yr, based on estimates from Niemi et al. (2005), where modelled landsliding rates of between 2.85-3.85 mm/yr were found to best fit measured 10Be concentrations (where bedrock weathering rates make up a small fraction, ~0.15 mm/yr, of the total erosion rate). The rest of the catchment is assumed to be eroding at our background/catchment-averaged erosion rate. This seems like a reasonable compromise, where the small catchment used in Niemi et al. (2005) is undergoing higher total erosion rates (in excess of 3 mm/yr as located in the Higher Himalaya) than we would expect in a much larger trans-Himalayan catchment.

As predicted, this doesn't make any significant changes to our results. In the instance where we have no system buffering, we do find that our outlet concentrations are much lower and the effect of increasing the background erosion rate across the rest of the catchment also has less impact of the outlet concentration. This suggests that the landslide derived material is dominating the outlet concentration. Again, the volumetric sediment flux estimates are systematically higher than the CRN-derived fluxes. We have also included same results in part b of this figure for average landslide erosion rates of 2 and 4 mm/yr (as labelled) to demonstrate the sensitivity of the results to this parameter.

[Figure]

When we include buffering of the landslide derived sediment in the system, the revised figure is nearly identical to the previous version:

[Figure]

Minor comments:
- p.3, l. 1-11: This would be a paragraph to also mention other studies that observed variability in the 10Be signal through repeated sampling such as in Godard et al., (2012) along the Marsyangdi.

Agreed – added in additional references.

- p.7, l. 20: within the variability of modern samples?

Yes, changed to variability.

I am looking forward seeing the final version published!

Maarten Lupker – ETH Zürich

**Reviewer 3 (Amanda Schmidt):**

This paper considers variability in in situ 10Be concentration in one catchment draining most of the central Himalayas in order to understand how temporal variability in 10Be concentration may affect estimated erosion rates. Overall I think the paper is an interesting contribution to add to empirical work previously done by Lupker and Gonzalez and modeling done by Niemi and Yanites. I have a few concerns about the paper that I think the authors could address fairly easily to make the paper acceptable for publication.

1) The authors have completely ignored the Earth Surface Processes and Landforms paper by Gonzalez that came out last year. In that paper Gonzalez and coauthors (full disclosure that I am the corresponding author on that paper) compare 10Be (both in situ and meteoric) measurements of pairs of samples from western China and also do a meta-analysis of previously published replicate studies. Not including this paper is a major oversight because it means that it feels like Dingle is reinventing the wheel rather than engaging in conversation with other manuscripts considering similar topics. In particular, I think that considering similarities and differences to the Gonzalez et al dataset would be appropriate for the discussion and the legwork Gonzalez did in a global meta-analysis should contribute to the introduction.

Many thanks for bringing this paper to our attention (this paper only came out a couple of months before our initial manuscript submission). We have added in additional text to our discussion (p.14 l.16- 21):

2) I am bothered by the way that the manuscript is structured with respect to the landslide modeling. The manuscript has a standard set up of introduction, methods, results, discussion, conclusion for the 10Be concentrations and derived denudation rates. However, the authors have thrown into the discussion section all of the background (introduction), methods, results, and discussion for their landslide model. I think that it would be more appropriate to introduce the model early, develop it in the methods, present the results in the results section, and discuss it in the discussion section.

As we discussed in our initial response to reviewers 1&2, we have played around with the structure of the paper and decided to keep with our initial structure. In order to introduce the modelling aspect, we would need to justify why we have undertaken it. The motivation for the modelling was based on the surprising results from the CRN analysis.

3) Although the authors do an excellent job of addressing a number of possible reasons for the variability in isotopic data, they do not (as far as I can tell) mention the one process that Lupker concluded dominated in the Himalayas – varying sourcing from catchments with similar average erosion rates but different production rates due to varying elevation.

Yes this is a good point. We have indirectly assessed this in Figure 8c, where we change the production rate of the area undergoing landsliding – we can see that under lower erosion rates, this has a much greater impact on outlet 10Be concentrations. The factors we later go on to discuss (e.g. storage/buffering) likely limit this impact however.

4) I'm confused about why you use concentration differences when you have location differences for your samples. It seems like in that case it is better to compare denudation rates (as Schaller did) rather than comparing concentrations, which are biased by variable upstream elevation. I agree that if you are comparing two samples at the exact sample (or nearly exact same) sample site, then comparing concentration is more appropriate, but given the variability in location, I think that comparing denudation may be better.

As we discussed in our initial responses to reviewers 1&2, we have chosen to compare concentrations for a couple of reasons. Firstly, in order to generate denudation rates, we are immediately imposing a series of assumptions onto our data. Secondly, many of our samples are taken in a relatively small geographical area relative to the full size of the catchment. Admittedly, a number of samples are spread upstream/downstream of the Alaknanda-Bhagirati confluence, but as shown in Figure 6 (which shows denudation rate) we see little variation in denudation rate between samples as well.

5) I would like to see some conclusions that you can draw that may help others to sample better. It's all well and good to say "be careful, the system doesn't work like you assume" but unless you can offer concrete suggestions, people will continue to happily run around and grab bags of sand and assume that everything is well averaged. What can you recommend we do?

Yes, this is an excellent point! We have added in a couple of statements and some extra analysis to fully explore this:

p.14 l.22 – p.15 l.8 and Figure 12

"Using the approximate range of concentrations documented at the Ganga outlet (5,000-30,000 atoms g$^{-1}$) as an example of natural variability, we can statistically constrain the number of samples required to capture this variability with repeat sampling. We proceed as follow: we produce a population of concentrations by choosing, at random, $x$ values from a Gaussian distribution with a mean of 17,500 atoms g$^{-1}$ and a standard deviation of 4,000 atoms g$^{-1}$, based on the values from the Ganga River samples. We repeat this procedure 100 times for each value of $x$, with $x$ (the number of samples in a population) varying between 3 and 50. If we assume that the standard deviation of the concentrations for each population is a proxy for concentration variability within a set of samples, then the mean standard deviation of the 100 populations for a given number of samples $x$, and the standard deviation around this mean, give an indication as to whether the variability is well constrained. This is exemplified in Figure 12: with increasing number of samples $x$ within a population, the mean standard deviation increases and converges asymptotically towards the true value of 4,000 atoms g$^{-1}$.

The standard deviation around the mean for the 100 populations generated for each number of samples $x$ (error bars on the figure) reduces with increasing sample number, i.e. the variability becomes better constrained. With 18 samples, the mean standard deviation is within 10% of the true standard deviation; more importantly, increasing the number of samples beyond 18 leads to minimal improvement, with the mean increasing by less than 0.3% per additional sample. We therefore suggest that 18 samples represent a good balance between cost and performance when trying to characterise the natural $^{10}$Be concentration variability of a river system similar to the Ganga River. It is important however to note that the error bars around the mean standard deviation are large. Even with 50 samples, 68% of the concentration populations (within one standard deviation of the mean assuming a Gaussian distribution of values – error bars on figure) will have a standard deviation within 23% of the true value (in the range ~3,100-4,500 atoms g$^{-1}$); nearly a third of the populations will therefore have a standard deviation beyond this bound. This figure is 35% and 44 % for 18 and 5 samples, respectively (with standard deviations of 3,610 ±1,020 and 3,000±1,250 atoms g$^{-1}$, respectively). These numbers may be influenced by the shape of the concentration distribution."

[Figure]

"Future sampling strategies in large Himalayan catchments should seek to incorporate multiple samples in both monsoon and non-monsoon conditions to better characterise temporal variability in $^{10}$Be concentrations"

In addition to these major points, I have a few minor comments:

Term use throughout: Why do you use CRN instead of 10Be? You only measured in situ 10Be, so it seems silly to refer to the generic term CRN throughout.

This is a fair point – we have largely replaced CRN with $^{10}$Be.

Acronyms throughout: Be sure to define all acronyms (CRN, CAIRN, CRONUS) at first appearance.

Thanks – added.

P5 L26: I would put the grain size you field sieved to here.

We didn't sieve any samples in the field – several kg of sediment was brought back to the UK where all of the sieving was then undertaken.

P6 L7: Can you justify that sometimes including this finer grain size doesn't affect your concentration measurements?

There is a nice section in Lupker et al. (2012) considering grain size bias in measured concentrations in the Himalaya. We have added a sentence or two to explain:

"While previous studies have demonstrated that different sediment grain size fractions may be selectively enriched in $^{10}$Be (e.g. Puchol et al., 2014; Schilgden et al., 2016), analysis from Lupker et al. (2012) on the 125-250 and 250-400 µm grain size fractions (from the same samples) at the Himalayan mountain front reveal no systematic differences in $^{10}$Be concentration."

Section 3.2: I think it would be extremely helpful to have a table that has the quartz amount, carrier mass, and AMS measurements (ratios, blanks, blank-corrected ratios) so that calculations can be checked and replicated. These are standard supporting tables these days. This is especially important since your blank values range from 4 to 54% of your measured ratios.

This has been added into the Appendix

Section 3.3: Be sure to include all the parameters that one needs to replicate your work (like assumed sediment density). It is also useful to include a table that one can easily input into an erosion rate calculator should they want to do a meta-analysis or recalculate your denudation rates as parameters change.

Density added into this section

Figures: All figures have text that is hard to read. I assume it is a reproduction for review issue, but wanted to ensure that you check that text is easily legible in final publication format.

Thanks for pointing this out.

Figure 2: If you aren't using lat/long, you need to say what datum and projection your coordinates are from. Given that Table 1 has lat/long in decimal degrees, you may consider changing figure 2 to use decimal degrees as well.

This is explained in the caption for Figure 1 ("Coordinates are projected in UTM Zone 44N"). Kept as Lat/Lon in Table 1 as this seems to be more standard for the online calculators if anyone wanted to reproduce.

Table 2: Why mean elevation instead of effective elevation? (And, I am assuming "average" is "mean" and not "median" or "mode". Probably would be good to clarify.)

Thanks – changed to mean elevation. This was included in the table to illustrate the differences/range in catchment elevation (and therefore possible differences in 10Be production rate) between these sub-catchments.

This review is intended to be constructive and I would be happy to clarify or answer questions for the authors.

Amanda Schmidt

[revised manuscript text omitted]

10  Puchol et al., 2014; Schildgen et al., 2016), analysis from Lupker et al. (2012) on the 125-250 and 250-400 $\mu$m grain size fractions (from the same sample) at the Himalayan mountain front reveal no systematic differences in [10]Be concentration. Following this procedure, samples were put through repeated dissolutions in aqua regia and diluted HF and $HNO_3$ solutions to remove mineral phases other than quartz. Quartz samples were then etched with HF to remove between 30 and 50 % of their volume. The purity of the clean quartz cores were then tested by ICP-OES. All the Al concentrations in the quartz cores were

15  below 300 ppm. Between 7 and 30 g of quartz cores were dissolved in concentrated HF. Samples were spiked with c. 220 $\mu$g of a [9]Be carrier produced in the cosmogenic isotope analysis facility at the Scottish Universities Environmental Research Centre (SUERC) from phenakite crystals. The [10]Be carrier concentration is c. 9 $\times 10^{-16}$ [10]Be/[9]Be. A procedural blank was prepared together with each group of samples. Be was isolated from the solutions following routine column chemistry (Darvill et al., 2015). [10]Be/[9]Be ratios of the produced BeO targets were measured with the 5 MV Pelletron AMS at the SUERC (Xu et al.,

20  2010). [10]Be data were calibrated against the National Institute of Standards and Technology standard reference material NIST SRM 4325. The activity of NIST SRM 4325 corresponds to a nominal [10]Be/[9]Be ratio of 2.79 $\times 10^{-11}$ for a [10]Be half-life of 1.36 $\times 10^6$ years. The processed blank ratios ranged between 4 and 54 % of the sample [10]Be/[9]Be ratios (see Table A1 for details). The uncertainty of this correction is included in the stated standard uncertainties.

**3.3 Denudation rate calculations**

25  Catchment-averaged denudation rates were calculated for each sample using the CAIRN (Catchment-Averaged denudatIon Rates from cosmogenic Nuclides) method (Mudd et al., 2016), which estimates production and shielding factors on a pixel-by-pixel basis, rather than a catchment-averaged shielding factor as in more commonly used CRN analysis packages such as CRONUS  (Cosmic-Ray prOduced NUclide Systematics in Earth) (Balco et al., 2008). An average rock density of 2650 kg m$^{-3}$ was used (the default for CAIRN). 
[revised manuscript text omitted]

20  $^{10}$Be concentrations. By  increasing the average landslide erosion rate (relative to the catchment-average erosion rate applied across the rest of the catchment) in our analysis, we indirectly assess the importance of such effects.

We calculate 'volumetric sediment flux' by combining the flux derived from background erosion rates with the calculated landslide flux, and compared these to sediment flux estimates derived from the $^{10}$Be concentration at the catchment outlet

25  (which we term the 'CRN-derived sediment flux'). For a catchment eroding at a uniform rate ($\epsilon$ in mm yr$^{-1}$), the CRN-derived sediment flux is the product of the erosion rate, catchment area ($A$ in km$^2$) and average rock density ($\rho$ in kg m$^{-3}$).

In this analysis, we assume that sediment storage between the region affected by landslides and the outlet is small relative to the total sediment flux of the catchment. Unlike the eastern and western Himalaya, the central Himalaya (which is largely drained by tributaries of the Ganga River) is comparatively void of large valley fills (Blöthe and Korup, 2013), which is likely to

30  limit large volumes of sediment storage and sediment residence times. Recent modelling has also suggested that approximately 50 % of coarse material generated by post-seismic landsliding is evacuated within 5 to 25 years (Croissant et al., 2017). In our scenarios, we initially assume complete evacuation of material to the outlet within a year. We then run additional analysis where much smaller proportions of the event material are mixed into the fluvial network in this first year (3, 5, 10 and 20% of the event sediment). The default and range of values tested for each parameter in the analysis are shown in Table 3.

Based on the above calculations, our results suggest that increasing the average landslide depth results in a marked decrease in outlet $^{10}$Be concentration, most notably between depths of 0.5-3 m (Fig. 8a). This can be explained through the exponential decay in $^{10}$Be production rates in the upper 2 m of the landslide (Lal, 1991; Stone, 2000; Niedermann, 2002). This reduction in concentration is greatest under lower background erosion rates. Increasing background erosion rates from 0.2-2.0 mm yr$^{-1}$

5 also reduces the effect of landsliding on outlet $^{10}$Be concentrations (Fig. 3b). Under lower background erosion rate, landslide material represents a greater proportion of the total sediment flux, so the system has less capacity to buffer the landslide input and the $^{10}$Be concentration is more sensitive to deeper landslides. We also find that outlet $^{10}$Be concentrations are sensitive to the average landslide surface production rate. Where the average surface production rate of the landsliding is increased (e.g. comparable to that expected in high altitude sub-catchments of the Ganga - see Table 2), predicted outlet $^{10}$Be concentrations

10 also increase relative to scenarios with otherwise identical parameter values (Fig. 8c). Interestingly, we also find that volumetric sediment flux estimates are consistently higher than CRN-derived fluxes (Fig. 8d). Increasing background erosion rates increases both CRN-derived and volumetric sediment flux estimates, but increasing average landslide depth or landslide  $^{10}$Be production rate can reduce CRN-derived sediment flux estimates to a much greater degree than volumetric flux estimates.

 The average landslide erosion rate was increased to 3.0 mm yr$^{-1}$, based

15 on estimates in Niemi et al. (2005), to mimic the effects of faster erosion rates in regions more prone to landsliding and landscapes without steady-state concentration profiles. Niemi et al. (2005) ran a series of numerical modelling scenarios to explore the ratio of landslide to bedrock weathering (background) erosion rates needed to reproduce measured CRN erosion rates in the Khudi catchment in Nepal. The best fit model runs were found to have landslide erosion rates of 3.35 mm yr$^{-1}$. By applying a comparable value of 3.0 mm yr$^{-1}$ to our calculations, a reduction in the absolute values and range of outlet

20  $^{10}$Be concentrations is produced. The initial maximum outlet concentration of $\sim$70,000 (in Fig. 8a) is reduced to 12,000 atoms g$^{-1}$ under the lowest background erosion rate scenarios (Fig. 9a). This range of outlet  $^{10}$Be variability is more comparable to that observed at the Ganga outlet, although outlet concentrations appear less sensitive to background erosion rates applied across the rest of the catchment. 
[revised manuscript text omitted]
. This is consistent with previous work using repeat [10]Be samples from tectonically active watersheds in China, where it was concluded that replicability of data in these types of landscapes is likely to be poor, and that larger sample populations are needed to better represent upstream denudation rates (Gonzalez et al., 2017).Results from our study also support this finding, where we demonstrate that multiple samples are required to better characterise the temporal variability

25  in [10]Be concentrations at the Himalayan mountain front.

     Using the approximate range of concentrations documented at the Ganga outlet (5,000-30,000 atoms g$^{-1}$) as an example of natural variability, we can statistically constrain the number of samples required to capture this variability with repeat sampling. We proceed as follow: we produce a population of concentrations by choosing, at random, $x$ values from a Gaussian distribution with a mean of 17,500 atoms g$^{-1}$ and a standard deviation of 4,000 atoms g$^{-1}$, based on the values from the

30  Ganga River samples. We repeat this procedure 100 times for each value of $x$, with $x$ (the number of samples in a population) varying between 3 and 50. If we assume that the standard deviation of the concentrations for each population is a proxy for concentration variability within a set of samples, then the mean standard deviation of the 100 populations for a given number of samples $x$, and the standard deviation around this mean, give an indication as to whether the variability is well constrained. This

is exemplified in Figure 12: with increasing number of samples $x$ within a population, the mean standard deviation increases and converges asymptotically towards the true value of 4,000 atoms g$^{-1}$.

The standard deviation around the mean for the 100 populations generated for each number of samples $x$ (error bars on the figure) reduces with increasing sample number, i.e. the variability becomes better constrained. With 18 samples, the mean standard deviation is within 10% of the true standard deviation; more importantly, increasing the number of samples beyond 18 leads to minimal improvement, with the mean increasing by less than 0.3% per additional sample. We therefore suggest that 18 samples represent a good balance between cost and performance when trying to characterise the natural $^{10}$Be concentration variability of a river system similar to the Ganga River. It is important however to note that the error bars around the mean standard deviation are large. Even with 50 samples, 68% of the concentration populations (within one standard deviation of the mean assuming a Gaussian distribution of values - error bars on figure) will have a standard deviation within 23% of the true value (in the range ∼3,100-4,500 atoms g$^{-1}$); nearly a third of the populations will therefore have a standard deviation beyond this bound. This figure is 35% and 44% for 18 and 5 samples, respectively (with standard deviations of ∼3,610±1,020 and 3,000±1,250 atoms g$^{-1}$, respectively). These numbers may be influenced by the shape of the concentration distribution.

[revised manuscript text omitted]

20    topographic relief, spatially variable climate and multiple geomorphic process domains, the use of $^{10}$Be concentrations to generate sediment flux estimates may not be truly representative, as comparable mean catchment  $^{10}$Be concentrations can be derived through dramatically different erosional processes. For a given  $^{10}$Be concentration, volumetric sediment flux estimates may vary and under certain conditions,  $^{10}$Be concentrations may under-estimate actual erosion rates and hence sediment flux. Future sampling strategies in large Himalayan catchments should seek to incorporate multiple samples in both

25    monsoon and non-monsoon conditions to better characterise temporal variability in $^{10}$Be concentrations.

[revised manuscript text omitted]

**Figure 9.** (a) Effect of  increasing average landslide  erosion rate to  3.0 mm yr$^{-1}$ on outlet  $^{10}$Be concentrations in response to varying landslide depths and catchment background erosion rates. The overall range in outlet concentrations is notably lower than in Fig. 8a. Increasing the catchment-averaged erosion rate only has an impact on outlet concentrations where the input of landslide material is smaller, suggesting that the outlet concentration is dominated by landslide-derived material. (b) Comparison of volumetric and CRN-derived sediment fluxes for the same model conditions, where marker colour corresponds to background erosion rate shown in part (a). The difference in volumetric and CRN-derived fluxes is much less than scenarios shown in Fig. 8d. In  general, the volumetric  flux is approximately double the CRN-derived  sediment flux. By increasing and decreasing the average landslide erosion rate to 4.0 and 2.0 mm yr$^{-1}$ as shown by the smaller black markers, this relationship varies slightly.

[Figure]

**Figure 10.** (a) Effect of event buffering on outlet  $^{10}$Be concentrations, where smaller fractions (3, 5, 10 and 20%) of the event sediment are mixed into the fluvial network based on two background erosion rates of 0.6 and 2.0 mm yr$^{-1}$ shown in blue and red, respectively. The event proportions are represented by the different dashed lines. The _average_ landslide surface  _erosion_ rate is set to  _3.0 mm_ yr$^{-1}$ . Under faster background erosion rates, the effect of larger landsliding events are more easily buffered in outlet  $^{10}$Be concentrations. (b) Comparison of volumetric and CRN-derived sediment fluxes for event buffering scenarios. Under these conditions, volumetric and CRN-derived sediment flux estimates are much more comparable.  _As the amount of landslide-derived material is mixed into the system increases_, volumetric _sediment_ fluxes  _become_ slightly larger  _than CRN-derived sediment fluxes_.

[Figure]

**Figure 11.** Volumetric sand (grain sizes <1 mm) proportions in sub-surface sediment samples along major tributaries of the Ganga River from Dingle et al., 2016.

[Figure]

**Figure 12.** Number of [10]Be samples required to capture the natural concentration variability of the Ganga River. Approximately 18 samples are required to be within 10% of the true standard deviation (or variability) of the system. Blue dots represent the mean standard deviation of 100 populations of concentrations for a given sample group size (between 3 and 50). Error bars represent the standard deviation of the mean standard deviation of those 100 populations, per sample group size. The solid horizontal red line represents the mean standard deviation value that the sample group sizes converge towards (4,000 atoms g[-1]). The two dashed red lines represent the number of samples required to be within 10% of the true standard deviation (labelled 90%), and the standard deviation expected from a set of five samples (labelled 78%).

[revised manuscript text omitted]

\*\* Details for this sample (BR924) are from Table 1 in Lupker et al. (2012). We have recalculated the erosion rate using the CAIRN method (Mudd et al., 2016).

**Table 2.** Catchment area,  mean catchment elevation and average [10]Be surface production rate for sub-catchments in the Ganga catchment

|  | Catchment area (km$^2$) |  Mean catchment elevation (m) | Surface production rate (atoms g$^{-1}$ yr$^{-1}$) |
|---|---|---|---|
| Sub-catchment 1 | 1,955 | 1,606 | 11.08 |
| Sub-catchment 2 | 4,635 | 4,716 | 56.02 |
| Sub-catchment 3 | 1,801 | 5,033 | 70.51 |
| Sub-catchment 4 | 1,449 | 1,642 | 24.28 |
| Sub-catchment 5 | 169 | 4,483 | 49.13 |
| Sub-catchment 6 | 181 | 1,868 | 12.82 |
| Sub-catchment 7 | 253 | 1,404 | 9.57 |
| Sub-catchment 8* | 39 | 4,806 | 49.61 |
| Ganga (whole) | 23,038 | 3,560 | 33.16 |

*This sub-catchment represents the area upstream of Kedarnath during the 2013 Alaknanda flooding

**Table 3.** Default and range of parameter values used in numerical analysis

| Parameter | Default value | Range of modelled values |
|---|---|---|
| Landslide depth (m) | 2 | 0.5 - 5.0 |
| Catchment area ($km^2$) | 23,000 | - |
| % of catchment impacted by landsliding | 0.5 | - |
| Catchment-averaged surface production rate (atoms $g^{-1}$ $yr^{-1}$) | 35 | - |
| Background erosion rate (mm $yr^{-1}$) | 0.5 | 0.2-2.0 |
| Landslide surface production rate (atoms $g^{-1}$ $yr^{-1}$) | 35 | 10-60 |
| Proportion of event sediment mixed into fluvial network (%) | 100 | 3-20 |

*This sub-catchment represents the area upstream of Kedarnath during the 2013 Alaknanda flooding